# Colorectal cancer liver metastatic growth depends on PAD4-driven citrullination of the extracellular matrix

A.E. Yuzhalin[1], A.N. Gordon-Weeks [2], M.L. Tognoli[1], K. Jones[1], B. Markelc [1], R. Konietzny[3], R. Fischer [3], A. Muth[4], E. O'Neill [1], P.R. Thompson[4], P.J. Venables[5], B.M. Kessler [3], S.Y. Lim[1,6] & R.J. Muschel[1]

Citrullination of proteins, a post-translational conversion of arginine residues to citrulline, is recognized in rheumatoid arthritis, but largely undocumented in cancer. Here we show that citrullination of the extracellular matrix by cancer cell derived peptidylarginine deiminase 4 (PAD4) is essential for the growth of liver metastases from colorectal cancer (CRC). Using proteomics, we demonstrate that liver metastases exhibit higher levels of citrullination and PAD4 than unaffected liver, primary CRC or adjacent colonic mucosa. Functional significance for citrullination in metastatic growth is evident in murine models where inhibition of citrullination substantially reduces liver metastatic burden. Additionally, citrullination of a key matrix component collagen type I promotes greater adhesion and decreased migration of CRC cells along with increased expression of characteristic epithelial markers, suggesting a role for citrullination in promoting mesenchymal-to-epithelial transition and liver metastasis. Overall, our study reveals the potential for PAD4-dependant citrullination to drive the progression of CRC liver metastasis.

[1] CRUK/MRC Oxford Institute for Radiation Oncology, Department of Oncology, University of Oxford, Old Road Campus Research Building, Roosevelt Drive, Oxford OX3 7DQ, UK. [2] Nuffield Department of Surgical Sciences, University of Oxford, John Radcliffe Hospital, Oxford OX3 9DU, UK. [3] Target Discovery Institute, Nuffield Department of Medicine, University of Oxford, Roosevelt Drive, Headington, Oxford OX3 7FZ, UK. [4] Department of Biochemistry and Molecular Pharmacology, University of Massachusetts Medical School, Worcester, MA 01605, USA. [5] Kennedy Institute of Rheumatology, Nuffield Department of Orthopedics, Rheumatology and Musculoskeletal Sciences, University of Oxford, Oxford OX3 7FY, UK. [6] Present address: Department of Biomedical Sciences, Faculty of Medicine and Health Sciences, Macquarie University, Sydney, Australia. These authors contributed equally: S.Y. Lim, R. J. Muschel.  Correspondence and requests for materials should be addressed to A.E.Y. (email: arseniy.yuzhalin@oncology.ox.ac.uk)

The extracellular matrix (ECM) in cancer is characterized as disorganized with increased production of some components (e.g., collagens) compared to normal tissue counterparts[1]. In addition to altered composition, cancer-associated ECM is subject to posttranslational modifications that alter the interactions of ECM proteins with cancer and stromal cells[2,3]. Certain ECM protein modifications are well described including proteolytic cleavage, glycosylation, and crosslinking[1]. For example, lysine crosslinking through oxidation increases the physical stiffness of the ECM in primary tumors, altering cancer cell signaling, enhancing epithelial-to-mesenchymal transition (EMT) and driving tumor progression[4].

Citrullination, the deimination of arginine residues to form peptides containing the noncoding amino acid citrulline is a well-recognized characteristic of chronic inflammation, as demonstrated in autoimmunity[5]. ECM proteins such as collagens are extensively citrullinated in patients with rheumatoid arthritis, leading to the generation of neoantigens and as a result the formation of autoantibodies[6]. In cancer, citrullination has been mostly undocumented, yet inflammation is one of the hallmarks of this disease[7] and so citrullination may play a role in cancer progression.

In inflammatory conditions, citrullination is catalyzed by peptidylarginine deiminases (PADs) 2 and 4[8]. In particular, citrullination of transcription factors and histones serves as a mechanism for transcriptional regulation of inflammatory mediators[9,10]. Furthermore, PAD4-mediated citrullination of histones also promotes chromatin decondensation which contributes to neutrophil extracellular trap (NET) formation[11].

Here, using high-throughput quantitative proteomics, we reveal that the ECM in liver metastases contains a higher proportion of citrullinated proteins than unaffected liver or primary colorectal carcinomas (CRC). Additionally, the liver metastases display higher levels of PAD4 than normal liver and primary CRC or adjacent colonic mucosa. Effects of citrullination are demonstrated in vitro with CRC cells grown on citrullinated collagen type I exhibiting an epithelial phenotype compared to those grown on unmodified collagen showing that ECM citrullination may promote mesenchymal-to-epithelial transition (MET). MET has been previously shown to be required for the successful growth of tumors at metastatic sites[12]. Pharmacological inhibition of PADs in vivo alters the balance toward increased mesenchymal markers in liver metastases and reduces metastatic growth. These results point to a new mechanism whereby cancer cells modulate ECM citrullination to promote liver metastatic progression.

## Results

**Composition of the matrisome from CRC liver metastases.** We first characterized the composition of the ECM (so called matrisome) from liver metastases. We modified the method of Naba et al.[13] to isolate ECM from murine experimental liver metastases by tissue decellularization followed by removal of contaminating cellular components through subcellular fractionation, and enzymatic depletion of nucleic acids and oligosaccharides (Supplementary Fig. 1A). The decellularized liver scaffolds retained a morphology consistent with ECM and contained characteristic ECM proteins including collagens, Fibronectin, and laminins (Supplementary Fig. 1B). The increased proportion of high molecular weight proteins in the preparations was consistent with enrichment of ECM proteins, as the ECM is composed of proteins of higher molecular weight than the intracellular compartment (Supplementary Fig. 1C).

This method was used to prepare ECM from human CRC liver metastases and adjacent uninvolved liver directly after surgical

resection for mass-spectrometry proteomic analysis (Supplementary Fig. 2A–C). Of 1097 proteins identified by qualitative analysis (Supplementary Data 1A), 158 were recognized as ECM components based on the Human Matrisome dataset[13] (Supplementary Fig. 3A, Supplementary Data 1B). Proteomics results exhibited low inter-sample variability in protein content and considerable overlap of the matrisomal proteins (Supplementary Fig. 3B, C). Matrisomal proteins found in hepatic metastases are listed in Supplementary Fig. 3D and Supplementary Data 1C. Interestingly, many of the identified proteins were initially not categorized as ECM-derived using the Human Matrisome dataset, but reclassification with the Panther system[14] identified the majority as ECM or components of extracellular vesicles (EVs) (Supplementary Fig. 3E, Supplementary Data 1D). This suggests that many of these non-matrisomal proteins could not be cellular contaminants, but instead were the result of retention of EV contents by the ECM.

We then applied quantitative label-free analysis to the proteomic data (Supplementary Data 2A, B). Using principal component analysis and clustering on the quantitative data (Fig. 1a, b) we identified 287 proteins with statistically significant differential abundance between the normal non-cancerous liver and metastases (two-way ANOVA, FDR-adjusted $P < 0.05$) (Fig. 1c), of which 69 with fold change of >3 were consistently up- or downregulated within the ECM (Fig. 1d, Supplementary Data 2 C, D). Many of the upregulated proteins have previously been linked to metastatic progression, including Versican, TIMP1, LTBP1-3, DDR1, and S100A10[15–19].

To confirm the proteomics, the expression of six selected proteins was assessed using immunoblotting and immunostaining on an independent set of resected hepatic metastases from additional CRC patients. Differential expression similar to the proteomic results was confirmed for all the proteins tested; versican, tenascin, S100A11, and COL5A1 were overexpressed, while matrix metalloproteinase-23 and mimecan were downregulated in the liver metastases compared to adjacent liver (Supplementary Fig. 4A–D).

**ECM proteins are highly citrullinated in liver metastases.** We next compared the citrullinome (i.e., the ensemble of citrullinated proteins) in liver metastases and adjacent normal liver using a PEAKS® software pipeline[20] to search for citrullinated residues in the data described above (Fig. 2a, Supplementary Fig. 5A). ECM from CRC liver metastases contained higher numbers of citrullinated peptides than adjacent normal liver (Fig. 2b, Supplementary Data 3A, B). Most of the citrullinated peptides were from proteins classified as matrisomal. Surprisingly, some proteins frequently citrullinated in rheumatoid arthritis[22] (e.g., vimentin, fibrin, and tenascin) were mainly unmodified in CRC liver metastases. On the other hand, proteins previously not reported to be citrullinated, for example, fibrillin-1 and emilin-1, were highly citrullinated in metastatic lesions (Supplementary Fig. 5B). To extend our results from human samples, we analyzed the ECM-associated citrullinome from murine experimental liver metastases generated through the intrasplenic injection of HT29 and LoVo cancer cells. There was considerable overlap with the data from human liver metastases, with fibrillin-1, emilin-1, and collagens being the most abundantly citrullinated proteins in liver metastases (Fig. 2c, d).

We then assessed the extent of overall protein citrullination in a different set of human liver metastases using two different methods, an ELISA and immunoblotting for 2,3-butanedione monoxime-modified citrullinated residues, both of which recognize citrullinated proteins, but not free citrulline[22]. These experiments confirmed the presence of increased amounts of

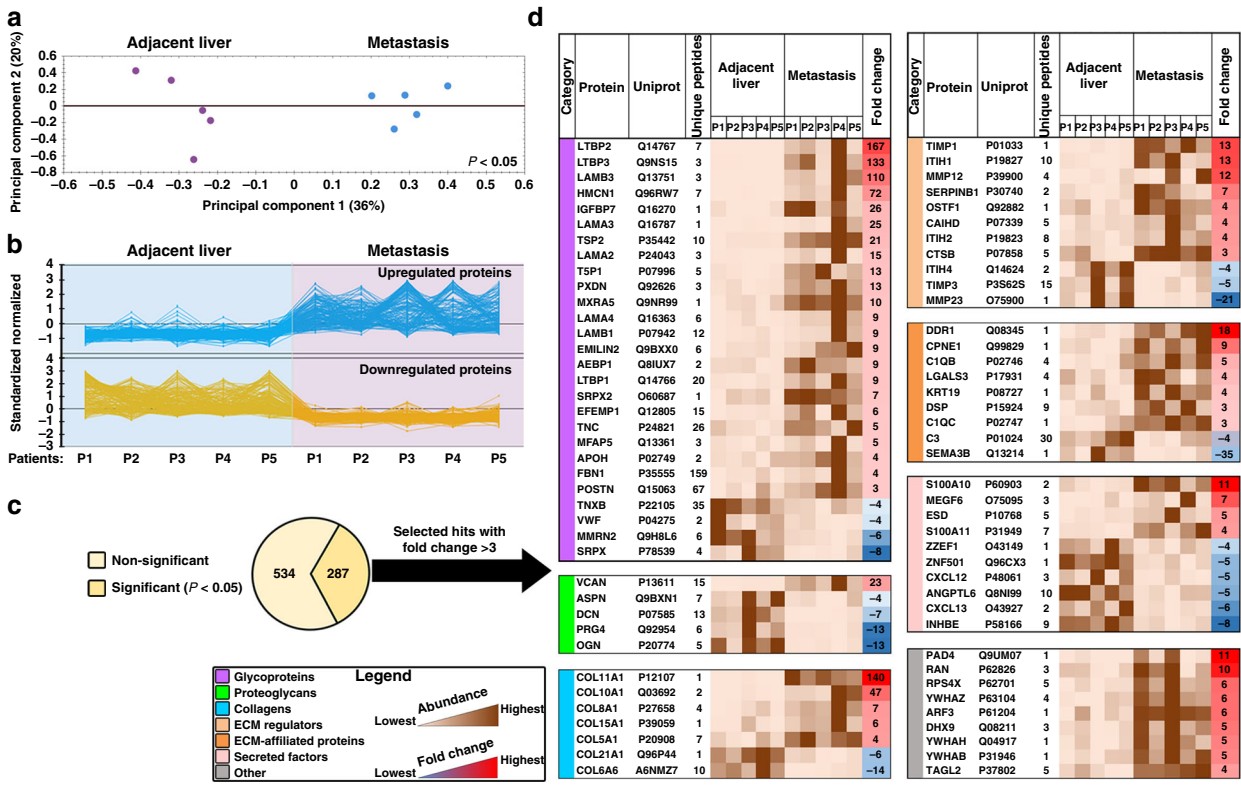

**Fig. 1** Label-free quantitative proteomics identifies the distinctive ECM composition of human CRC hepatic metastases. **a** Decellularized and ECM-enriched fractions from CRC hepatic metastases and paired adjacent unaffected liver tissues ($n = 5$ per group) were analyzed using LC-MS/MS with the following label-free quantitation. Shown is the principal component analysis of the relative protein abundance between metastasis and adjacent liver groups ($P < 0.05$). Percentage of variance is shown in parentheses. **b** Clustering analysis graph reveals categories of proteins differentially expressed in metastasis and adjacent liver groups. **c** Statistically significant differentially expressed proteins (287 out of 821 hits, two-way ANOVA, $P < 0.05$) were further classified either as ECM proteins or other proteins. **d** A heat map illustrating significantly different matrisome proteins (69 hits), which were selected based on a fold change threshold of > 3 and ranked in accordance with ECM category and fold change. The last gray colored table represents 9 selected other hits (i.e., non-ECM proteins yet significantly different in ECM-enriched samples). ECM extracellular matrix, P patient

citrullinated proteins in liver metastases compared to adjacent liver, primary CRC, or uninvolved colon samples (Fig. 2e, f, Supplementary Fig. 6A, B). The level of citrullinated peptides was also increased in ECM from CRC liver metastases compared to adjacent non-cancerous liver, primary CRCs, and uninvolved colon (Fig. 2g). Collectively, these data demonstrate that citrullinated ECM proteins are characteristic of liver metastases, but not primary colon tumors suggesting that citrullination may be an important component of the metastatic process.

**PAD4 is deposited in the ECM of hepatic metastases**. We then considered peptidylarginine deiminase 4 (PAD4) as a candidate to account for the increased citrullination of proteins in the liver metastatic ECM, because it was 11 times more abundant in the metastatic ECM than the uninvolved liver based on proteomics data (Fig. 1d). None of the other PADs were detected in the ECM by proteomics. PAD4 levels in lysates of human liver metastases showed a trend to be increased compared to the adjacent liver based on ELISA measurements and were significantly increased based on immunoblotting (Fig. 3a, b, Supplementary Fig. 6A, B). PAD4 exists as a 74 kDa monomer, which is less enzymatically active than dimeric PAD4[23]. The dimeric form was generally more abundant in our samples (Supplementary Fig. 6A, B). In addition, more minor intermediate bands which could include degradation products and some non-specific reactivity were seen (Supplementary Fig. 6A, B). These bands were enhanced after overexpression. Further the antibody shows specificity in that it reacts with recombinant PAD4 (Supplementary Fig. 6D). The

concentration of $Ca^{+2}$ which regulates dimerization is likely to vary depending on the nature of the sample[23]. We did not find increased levels of PAD4 in tissue lysates from human primary CRC or in adjacent uninvolved colon samples, suggesting that PAD4 upregulation is specific to liver metastases (Fig. 3a, b, Supplementary Fig. 6B).

To identify the cell types that produce PAD4 in the hepatic metastatic environment, we sorted cells from murine experimental liver metastases generated from three different human CRC cell lines into four groups: cancer cells expressing GFP, granulocytes (CD11b⁺Ly6G^hi), other myeloid cells (CD11b⁺Ly6G^lo) and the remaining stromal cells (GFP⁻, CD11b⁻, Ly6G⁻) (Supplementary Fig. 6C). Strikingly, for each tumor type, cancer cells expressed considerably higher amounts of PAD4 when compared with host cells (Fig. 3c). Further, PAD4 protein was produced by multiple cancer cell lines, including those isolated from metastatic sources (LoVo, SW620) and also found in experimental liver metastases (Supplementary Fig. 6D). To confirm that neutrophils do not produce the majority of PAD4 in metastatic microenvironment, we depleted neutrophils using Ly6-G antibody[24] (Supplementary Fig. 7A). Liver metastases from mice depleted of neutrophils had similar levels of PAD4 and protein citrullination as controls (Supplementary Fig. 7B, C). Thus, CRC cells often express PAD4 and appear to be the predominant source of PAD4 in the metastatic microenvironment in murine models.

We next investigated the mechanism through which cellular PAD4 is delivered to the ECM. PAD4 concentration increased

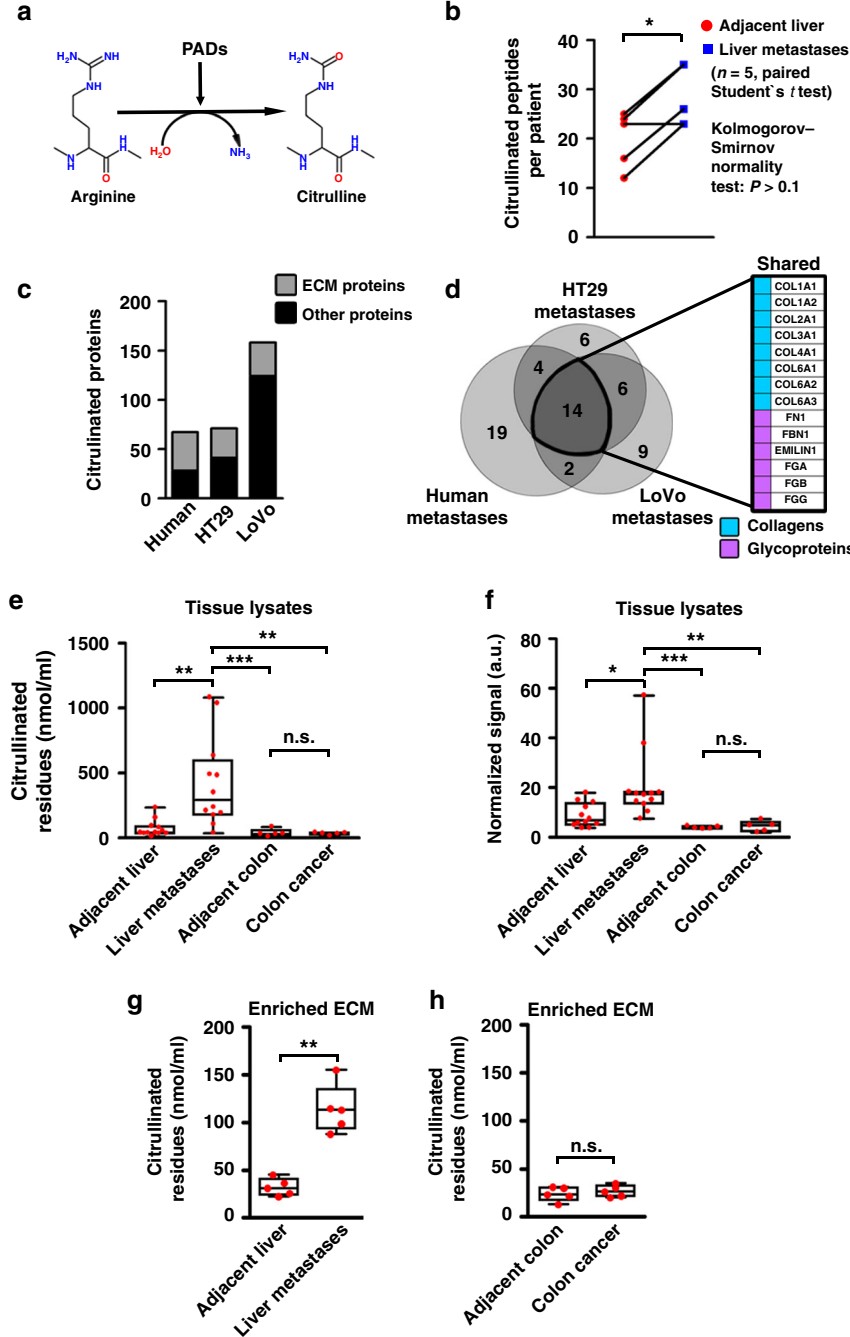

**Fig. 2** ECM proteins in human CRC hepatic metastases are extensively citrullinated. **a** Schematic of PAD-mediated modification of arginine to citrulline residues. **b** Quantification of citrullinated peptides identified by LC-MS/MS in the enriched ECM from CRC hepatic metastases and paired adjacent unaffected liver tissues (n = 5 per group). **c** Quantification of citrullinated proteins (categorized as ECM proteins or other proteins) identified in the ECM from human CRC liver metastasis tissues (n = 5) and ECM of experimental liver metastases generated using HT29 (n = 2 biological replicates) and LoVo (n = 2 biological replicates) CRC cells. **d** Venn diagram demonstrating an overlap of citrullinated ECM proteins identified in human and experimental CRC hepatic metastases from **c**. Shared proteins are listed in the box to the right. **e** ELISA for citrullinated proteins was performed on tissue lysates of human CRC hepatic metastases, paired adjacent unaffected liver tissues (n = 12 per group), human primary CRC lesions and paired adjacent unaffected colon tissues (n = 5 per group). Representative of two experiments. **f** Densitometry analysis of immunoblotting for citrullinated proteins performed on tissue lysates extracted from patients with CRC hepatic metastases and paired adjacent unaffected liver tissues (n = 12 per group) and patients with primary colon adenocarcinomas and paired adjacent unaffected colon tissues (n = 5). For the densitometry, the sum of all bands was used. Densitometry results were normalized to loading control. Representative of two experiments. **g**, **h** ELISA for citrullinated proteins was performed on decellularized and biochemically enriched ECM from human CRC hepatic metastases, paired adjacent unaffected liver tissues (n = 5 per group) (**g**), and human primary CRC lesions and paired adjacent unaffected colon tissues (n = 5 per group) (**h**). Experiment was performed once. Throughout, error bars indicate range, box bounds indicate second and third quartiles, center values indicate median. *P < 0.05, **P < 0.01, ***P < 0.001, n.s. = non-significant, paired Student's t test for **b** (P > 0.1 for Kolmogoro–Smirnov normality test), Mann–Whitney U test for **g** and **h**, and Kruskal–Wallis test with Dunn's post-test for **e** and **f**. ECM extracellular matrix

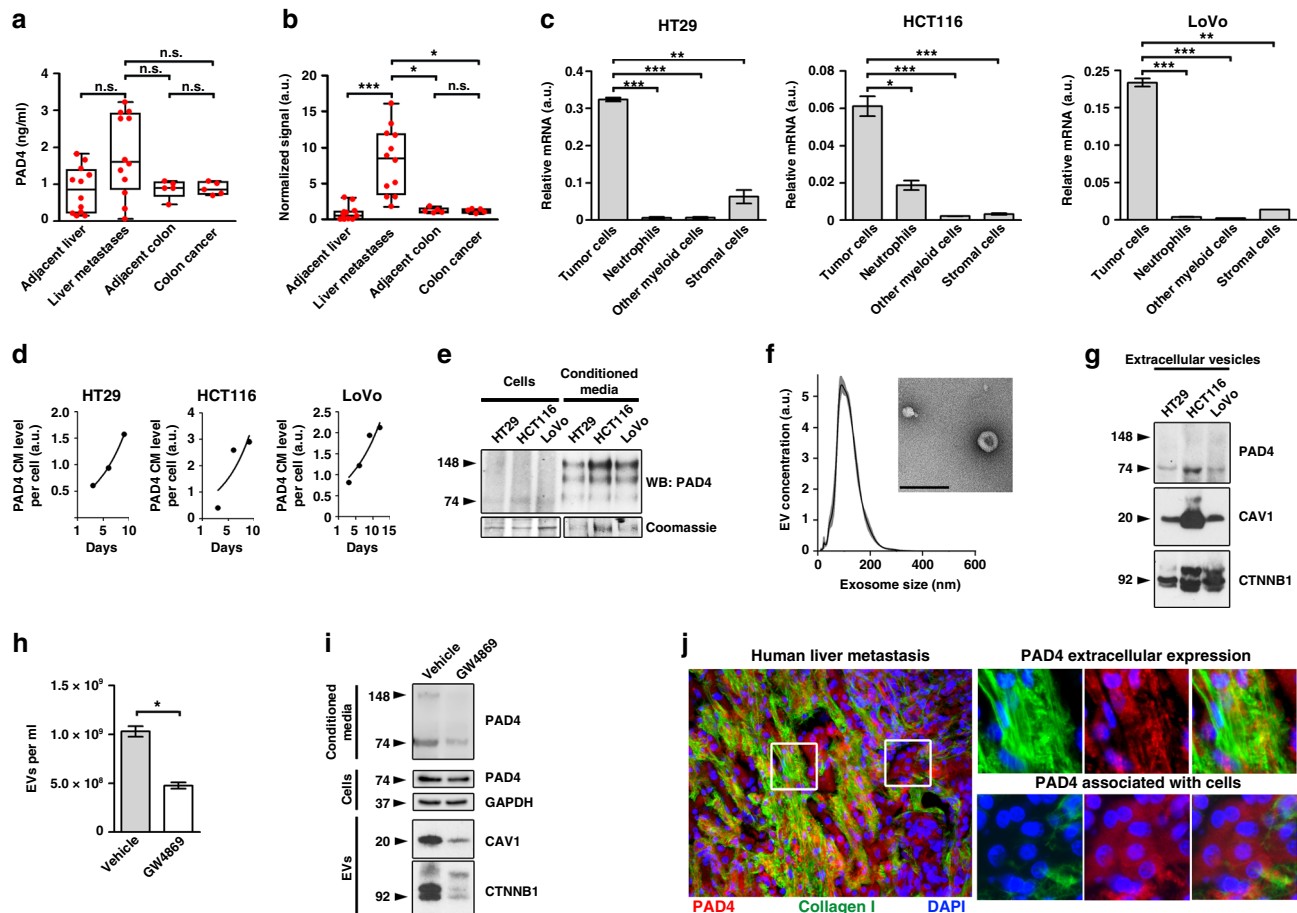

**Fig. 3** PAD4 is produced by CRC cells and is contained in EVs. **a** ELISA for PAD4 was performed on tissue lysates extracted from human CRC hepatic metastases, paired adjacent liver tissues ($n = 12$ per group), human CRC lesions and paired adjacent colon tissues ($n = 5$ per group). **b** Densitometric analysis of immunoblotting for PAD4 performed on tissue lysates of human with CRC hepatic metastases and paired adjacent unaffected liver tissues ($n = 12$) and human primary CRC lesions and paired adjacent unaffected colon tissues ($n = 5$). For the densitometry, the most expressed 148 kDa dimer band was measured. Densitometry was normalized to loading control. **c** Relative (to HPRT) mRNA levels of PAD4 in CRC cells, granulocytes, other myeloid cells and the stromal cells isolated from CRC experimental hepatic metastases ($n = 2$ biological replicates per group). **d** ELISA for PAD4 was performed on the conditioned media collected from cultured CRC cells at the indicated time points and normalized to the total cell number. **e** Immunoblotting for PAD4 was performed on HT29, HCT116 and LoVo cells and their corresponding conditioned media 6 days post-seeding. Coomassie stain was used as a loading control. **f** Nanoparticle tracking analysis of particle size distribution (graph) and electron microscopy (image) of EVs isolated from HT29 cells. Scale bar = 250 nm. **g** Immunoblotting for indicated proteins in EVs isolated from CRC cells. Representative of two experiments. **h** Nanoparticle tracking analysis of particle concentration of EVs collected from HT29 cells treated with vehicle or GW4869. **i** Immunoblotting for indicated proteins in cells, conditioned media and EVs from cultured HT29 cells treated with vehicle or GW4869. **j** Representative images of co-immunostaining for PAD4 and collagen I in human CRC hepatic metastasis tissue. Scale bar = 100 μm. For **a** and **b**, error bars indicate range, box bounds indicate second and third quartiles, center values indicate median, For **c** and **h**, error bars indicate s.e.m., center values indicate mean (*$P < 0.05$, **$P < 0.01$, ***$P < 0.001$, n.s. = non-significant, Mann–Whitney U test for (**h**), and Kruskal–Wallis test with Dunn's post-test for **a**–**c**). EV extracellular vesicles, CM conditioned media, WB immunoblotting

with time in CRC cell conditioned media, and extracellular molecule was present in its more active dimer form (Fig. 3d, e). Although this data demonstrates secretion of PAD4 by CRC cells, PAD4 lacks a defined secretory peptide raising the question of mechanisms for its release. We asked whether PAD4 was present in EVs, which would be consistent with the extensive deposition of proteins associated with EVs in the ECM of human liver metastases (Supplementary Fig. 3E). In keeping with deposition of PAD4 through vesicle release, EVs isolated from cancer cell conditioned media (Fig. 3f) contained PAD4 as well as many characteristic EV markers including caveolin-1 and β-catenin and lacked markers of ER and of Golgi apparatus (Fig. 3g, Supplementary Fig. 8A). PAD4 was enriched in EVs in the conditioned medium (Supplementary Fig. 8B). Exposure of HT29 to the sphingomyelinase inhibitor GW4869, a potent inhibitor of EV release[25], led to more than a twofold reduction in EV number

(Fig. 3h), and a corresponding decrease in PAD4 levels in the conditioned media, without altering intracellular PAD4 levels (Fig. 3i). HT29 but not HCT116 and LoVo, was unaffected by GW4869 in a WST-1 assay of cell proliferation and viability (Supplementary Fig. 8C). Hence EVs were only analyzed from HT29. Finally, immunohistochemistry of human CRC liver metastases revealed extracellular PAD4 localization. The extracellular PAD4 colocalized to collagens type I and IV providing additional evidence for the presence of PAD4 in the tumor ECM (Fig. 3j and Supplementary Fig. 8D). Additional PAD4 was associated with cellular areas.

Taken together, these data testify to the deposition of PAD4 in the ECM of hepatic metastases. EV-driven secretion of proteins without secretory peptides such as IL−1β has previously been documented[26], and while these data do not exclude alternative modes of secretion, they suggest that PAD4 can be released in EVs.

**Citrullinated collagen alters CRC adhesion, motility, and EMT.** To investigate the effect of ECM citrullination on cancer cell phenotypes, we plated CRC cells on citrullinated and non-citrullinated collagen type I. We chose collagen type I for these studies because it was one of the most highly expressed component of the liver metastatic ECM and also was extensively citrullinated (Supplementary Data 2B, Supplementary Fig. 5B). To generate citrullinated collagen, we incubated recombinant collagen type I with activated recombinant PAD4. As a control we added the PAD inhibitor BB-Cl-amidine[27] to inhibit citrullination (Fig. 4a, Supplementary Fig. 9A). Citrullinated collagen type I did not alter viability or proliferation of cultured CRC cells (Supplementary Fig. 9B, C). Adhesion of CRC cells was significantly greater to citrullinated collagen type I than to the non-citrullinated controls (Fig. 4b, Supplementary Fig. 10). Cell motility, measured by median velocity and track length was also significantly decreased in colon cancer cells plated on citrullinated

collagen type I compared to controls, albeit by only ~20–30% (Fig. 4c, d). These data implicate citrullination of the ECM in the promotion of cancer cell adhesion and inhibition of their motility, characteristic features of an epithelial cell phenotype.

We then examined the effect of cell adhesion to citrullinated collagen type I on downstream signaling. Focal adhesion kinase (FAK) is an integrin-dependent regulator of cell motility linked to EMT[28]. Phosphorylation of FAK and of its downstream targets, ERK and JNK, was decreased in CRC cells seeded onto citrullinated collagen type I in comparison to control, consistent with the increased attachment and decreased motility observed (Fig. 4e). Moreover, when plated on citrullinated collagen type I, the MC38 and LoVo CRC cells exhibited increased expression of epithelial markers (Fig. 4f, Supplementary Fig. 11). In particular, RNA expression of mesenchymal markers N-cadherin (*CDH2*) and Snail1 and 3 (*SNAI1* and *3*) were reduced in MC38 cells plated onto citrullinated collagen, while expression levels of

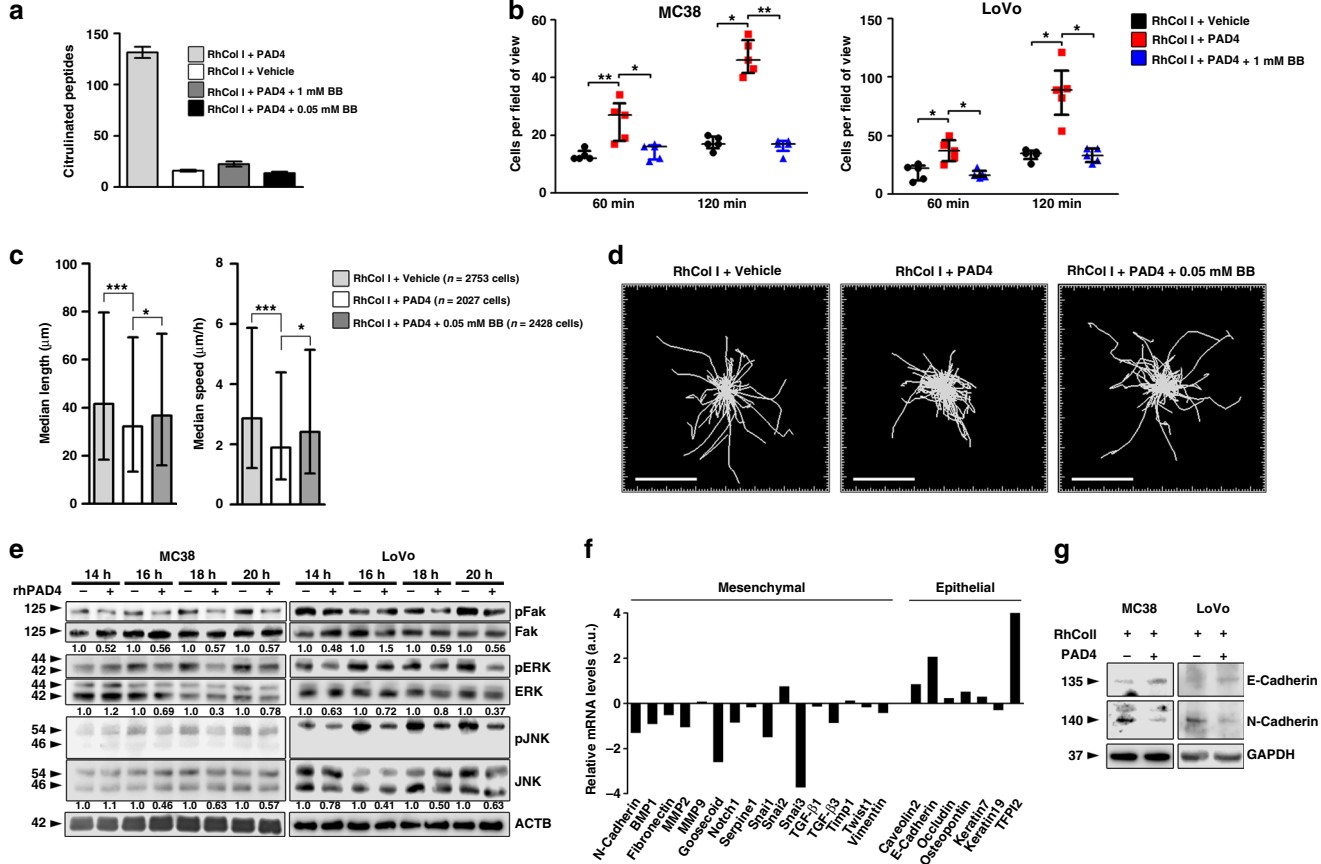

**Fig. 4** Citrullination of collagen I alters adhesion, motility and epithelial mesenchymal plasticity of CRC cells. **a** Recombinant collagen type I alone, collagen type I pre-treated with recombinant PAD4, or collagen I pre-treated with recombinant PAD4 and BB-Cl-amidine, were incubated and subjected to LC-MS/MS analysis. Shown are the numbers of citrullinated peptides. Repeated twice. **b** GFP+ CRC cells were plated in wells pre-coated with either collagen type I alone, collagen type I pre-treated with recombinant PAD4 or collagen type I pre-treated with PAD4 and BB-Cl-amidine. Cells were imaged at the indicated time points using an epifluorescence microscope. Four to five fields of view per condition were taken. Shown are the cell numbers attached to the plate for each condition. **c** GFP+ MC38 cells were plated in wells pre-coated with either collagen type I alone, collagen type I pre-treated with recombinant PAD4, or collagen type I pre-treated with addition of PAD4 and BB-Cl-amidine. Individual cells were tracked using time-lapse microscopy. Shown is the quantification of cell median length and speed. **d** Representative images of cell tracks from the experiment in (**c**). Scale bar = 200 μm. **e** Cancer cells were plated in wells pre-coated with either collagen type I alone or collagen type I pre-treated with recombinant PAD4. At indicated times cells were collected and probed for the indicated proteins. Numbers indicate relative expression. **f** Shown is mRNA expression of different EMT genes in MC38 cells seeded on collagen type I pre-treated with PAD4 compared to those seeded on unmodified collagen type I 18 h post-seeding. Shown is average of 2 technical replicates. **g** Immunoblotting for the indicated proteins performed on cell lysates collected from the experiment in (**f**). GAPDH was used as a loading control. For **a** and **b**, error bars indicate s.e.m. center values indicate mean (\*P < 0.05, \*\*P < 0.01, Kruskal–Wallis test with Dunn's multiple comparison post-test). For **c**, error bars indicate interquartile range, center values indicate median (\*P < 0.05, \*\*\*P < 0.001, Kruskal–Wallis test with Dunn's multiple comparison post-test). CTL control, BB BB-Cl-amidine

E-cadherin (*CDH1*) and other epithelial markers were increased. Reciprocal changes in N- and E-cadherin were confirmed at the protein level by immunoblotting (Fig. 4g). Thus, adhesion to citrullinated collagen type I promoted alterations in EMT markers in CRC cell lines.

**PAD inhibition abates liver metastatic growth and alters EMT**. To determine whether analogous events occur in vivo, we treated mice in an experimental liver metastasis experiment with the PAD inhibitor BB-Cl-amidine[27]. This resulted in a consistent decrease in the amount of citrullinated proteins in the liver colonies of the human CRC cells, although there was substantial variation between individual mice (Fig. 5a, b, c). Because the commercial antibody to 2,3-butanedione monoxime-modified citrulline is human-specific, we used immunohistochemistry for citrullinated Histone H3 to demonstrate a trend to reduced citrullination in the MC38 murine experimental metastasis model (Fig. 5d, e). The decrease in citrullinated proteins in the liver colonies correlated with a three- to fourfold reduction in metastatic burden (Fig. 5f–i). Metastases from untreated mice with more extensive citrullination exhibited strong expression of the epithelial markers E-cadherin, cytokeratin 7 and tight junction protein-1 and decreased expression of the mesenchymal markers ZEB1 and N-cadherin (Fig. 6a, b). In contrast, BB-Cl-amidine-treated tumors exhibited the reverse with decreased expression of epithelial markers and increased expression of mesenchymal markers. Thus, reduction of PAD activity and decreased citrullination enhanced expression of mesenchymal markers and concurrently impaired metastatic growth.

Administration of BB-Cl-amidine had no effect on growth of HCT116 subcutaneous xenografts but resulted in an approximately two- to threefold decrease in LoVo and MC38 xenograft volume (Fig. 6c). The markers of epithelial-mesenchymal plasticity of these subcutaneous tumors were not altered upon BB-Cl-amidine treatment (Fig. 6d), suggesting that PAD/ citrullination did not have a substantial effect on tumors in a non-metastatic subcutaneous location.

We then asked whether downregulation of the *PAD4* gene in CRC cells had analogous effects to pharmacological inhibition. *PAD4* levels were reduced by introduction of the shRNA in HT29 and HCT116 cells (Fig. 7a) and resulted in decreased intracellular and secreted protein citrullination (Fig. 7b) in comparison with cells transfected with control lentivirus. In tissue culture expression of the EMT markers N- and E-cadherin was unchanged suggesting that intracellular PAD4 and intracellular citrullination have minimal effects on epithelial-mesenchymal plasticity in these cells (Supplementary Fig. 12A). This is in striking contrast to the shift in EMT markers observed after plating the cells with downregulated PAD4 on citrullinated collagen type I suggesting that engagement with citrullinated ECM rather than intracellular PAD4 drove the changes in EMT characteristics. Of note, overexpression of *PAD4* following transduction by an expression vector did not change the amounts of N- or E-cadherin or Slug (Supplementary Fig. 12B).

Subcutaneous tumors from the PAD4-depleted HT29 and HCT116 cells had diminished PAD4 levels. Citrullination of proteins in cell lysates and decellularized and enriched matrices from subcutaneous xenografts was also decreased (Fig. 7c). HT29 cells had reduced growth in vitro, while HCT116 had a slight yet significantly reduced saturation density. Both showed decreased growth as subcutaneous tumors (Fig. 7d, e). We injected wild-type and *PAD4* knockdown cells intrasplenically to assess their growth as liver colonies. Strikingly, *PAD4*-deficient cells failed to form experimental liver metastases, suggesting the importance of this protein in the formation of liver colonies in vivo (Fig. 7f–h).

We next performed a spontaneous metastasis assay using orthotopic injection of HCT116 cancer cells into the cecal wall of mice (Fig. 8a). The cells with wild-type or *PAD4* gene knockdown led to bowel tumors with similar weights in contrast to the decreased growth at the subcutaneous site (Fig. 8b). The number of visible liver macrometastases (1–3 mm), however, was considerably decreased in the mice with *PAD4*-deficient cecal tumors (Fig. 8c). To evaluate the number of micrometastases, we sectioned the livers and counted the number of lesions as assessed by GFP positivity or histological evaluation (Fig. 8d). Both methods confirmed the marked reduction of micrometastases in the *PAD4* gene knockdown group compared to controls (Fig. 8e, f). Unexpectedly, the expression of *PAD4* was higher in metastases from mice injected with *PAD4*-deficient cells (Fig. 8g), even though *PAD4* levels were lower in the primary lesions of the *PAD4* knockdown group (Fig. 8h). This data implies that the few established metastases that developed may have arisen from cells which had escaped *PAD4* knockdown. This finding is consistent with a metastasis-specific requirement for PAD4. Taken together, these data suggest that reduction of in vivo activity of PAD4 in liver metastatic CRC cells diminishes growth of liver colonies.

## Discussion

ECM and its modifications orchestrate many facets of tumor biology including tumorigenesis, invasion and metastasis. The liver is the most common site for distant metastasis in CRC and poor survival for this disease is often attributable to metastasis[29]. Here we identify citrullination generated by PAD4 derived from tumor cells as a driver of human CRC liver metastases. Our experiments showing that citrullination of the ECM and expression of PAD4 facilitate human liver CRC metastasis may create opportunities for development of biomarkers and for therapeutic targeting.

We found PAD4 at higher levels in human CRC liver metastases than in primary CRC, or adjacent colon or liver. Consistent with the enzymatic activity of PAD4, citrullinated residues were present at higher levels in both total lysates and in the ECM of liver metastatic tissues than in primary CRC, liver, or colon. Our evidence suggests that extracellular PAD4, possibly through citrullination of collagen type I and other ECM proteins, alters the characteristics of CRC cells. Downregulation of PAD4 in tissue culture, a situation in which ECM is sparse did not alter EMT characteristics. However, plating cells on citrullinated collagen altered their EMT features regardless of intracellular PAD4. In experimental liver metastasis models, downregulation or pharmacological inhibition of PAD4 reduced citrullination and liver metastatic growth. Taken together, these experiments suggest that the citrullination of ECM molecules requiring extracellular PAD4 could be an important event in altering CRC cell signaling during metastatic growth. Unlike the CRC cells used here, downregulation of PAD4 in breast cancer cells[30] resulted in increased EMT markers Smad4 and p-Smad2, while overexpression of PAD4 in lung cancer cells[31] inhibited EMT through suppression of Elk1. Thus, signaling to reduce EMT marker expression may be mediated both intra- and extracellularly, reinforcing the possibility that PAD4 and citrullination might be a therapeutic target in liver metastases.

Although our work has focused upon PAD4, we recognize that citrullination can be catalyzed by other homologous PAD enzymes, including PADs 1–3 and 6. PAD2 and 4 are most broadly expressed in adults[8] and only PAD4 has a canonical nuclear localization sequence[32]. Of the PADs, however, PAD4 has been more often suggested to have the potential for involvement in cancer as it is overexpressed in tissues from many different malignancies including CRC[33], while PAD2 expression is downregulated in CRC

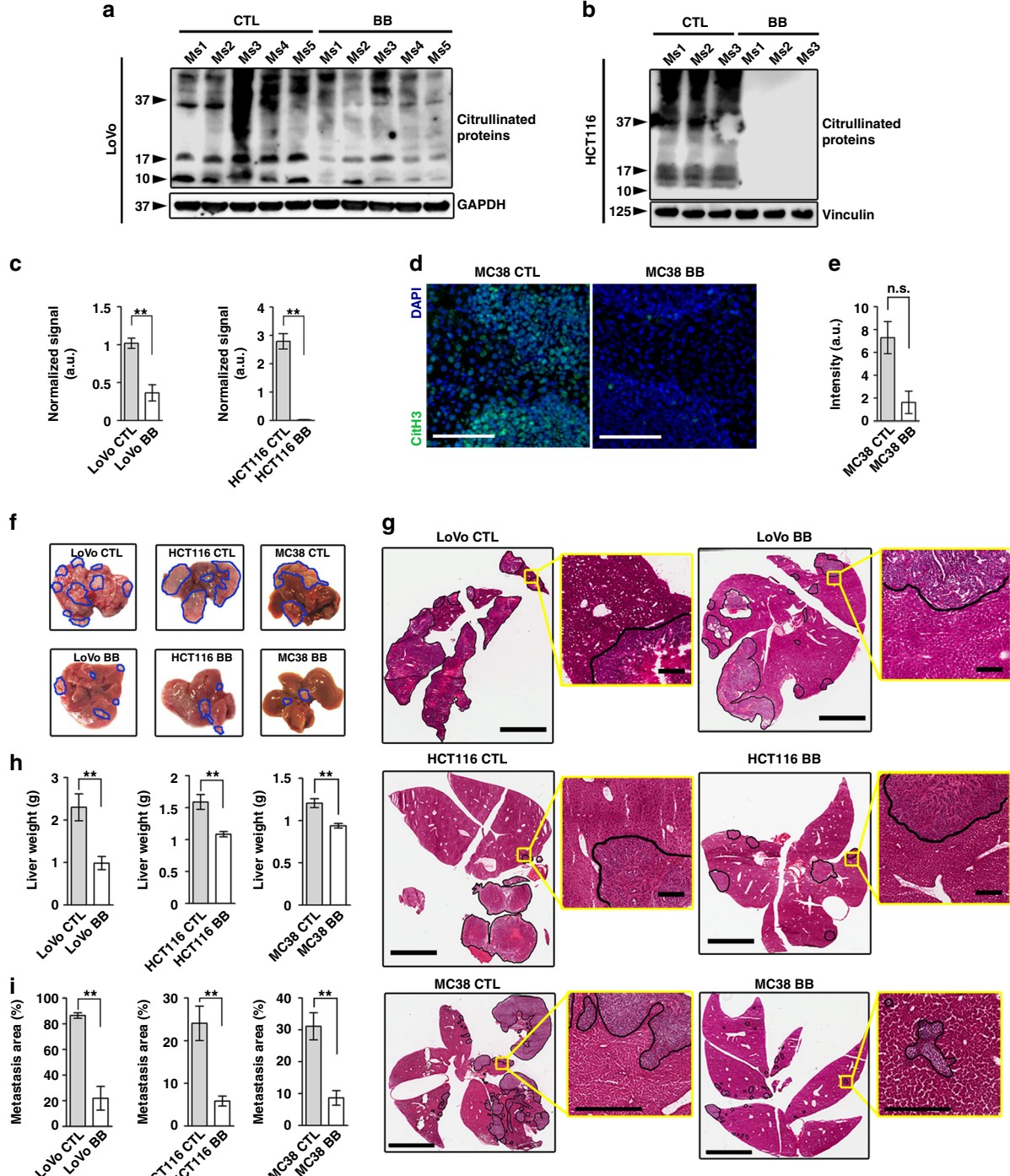

**Fig. 5** Pharmacological inhibition of PADs diminishes experimental hepatic metastases in vivo. **a, b** Immunoblotting for citrullinated proteins performed on tissue lysates extracted from experimental LoVo and HCT116 hepatic metastases from mice treated with vehicle or BB-Cl-amidine. Each lane indicates a biological replicate. GAPDH and Vinculin were used as a loading control. **c** Densitometric analysis for (**a**) and (**b**). **d** Representative images of immunostaining for citrullinated Histone H3 in experimental MC38 hepatic metastases from mice treated with vehicle ($n = 5$ biological replicates), or BB-Cl-amidine ($n = 6$ biological replicates). $n = 3$ sections per group, three images were analyzed per section. Scale bar = 100 μm. **e** Staining intensity quantification for (**d**). **f** Representative images of livers with experimental LoVo, HCT116 and MC38 hepatic metastases from mice treated with vehicle ($n = 7$, 6, and 5 biological replicates, respectively) or BB-Cl-amidine ($n = 8$, 6, and 6 biological replicates, respectively). Metastatic nodules are outlined in blue. Representative of two experiments. **g** Representative H&E-stained and scanned slides of dissected livers from the experiment in (**f**). Metastatic regions are outlined in black. Scale bar = 4 mm (scale bar of inserts = 200 μm). **h** Weight of excised metastasis-bearing livers from the experiment in (**f**). **i** Metastatic area based on the assessment of H&E-stained sections of livers from the experiment in (**f**). At least three different sections per mouse were analyzed. Throughout, error bars indicate s.e.m. center values indicate mean (**\*\*$P < 0.01$, n.s. = non-significant, Mann–Whitney U test). CTL control, BB BB-Cl-amidine

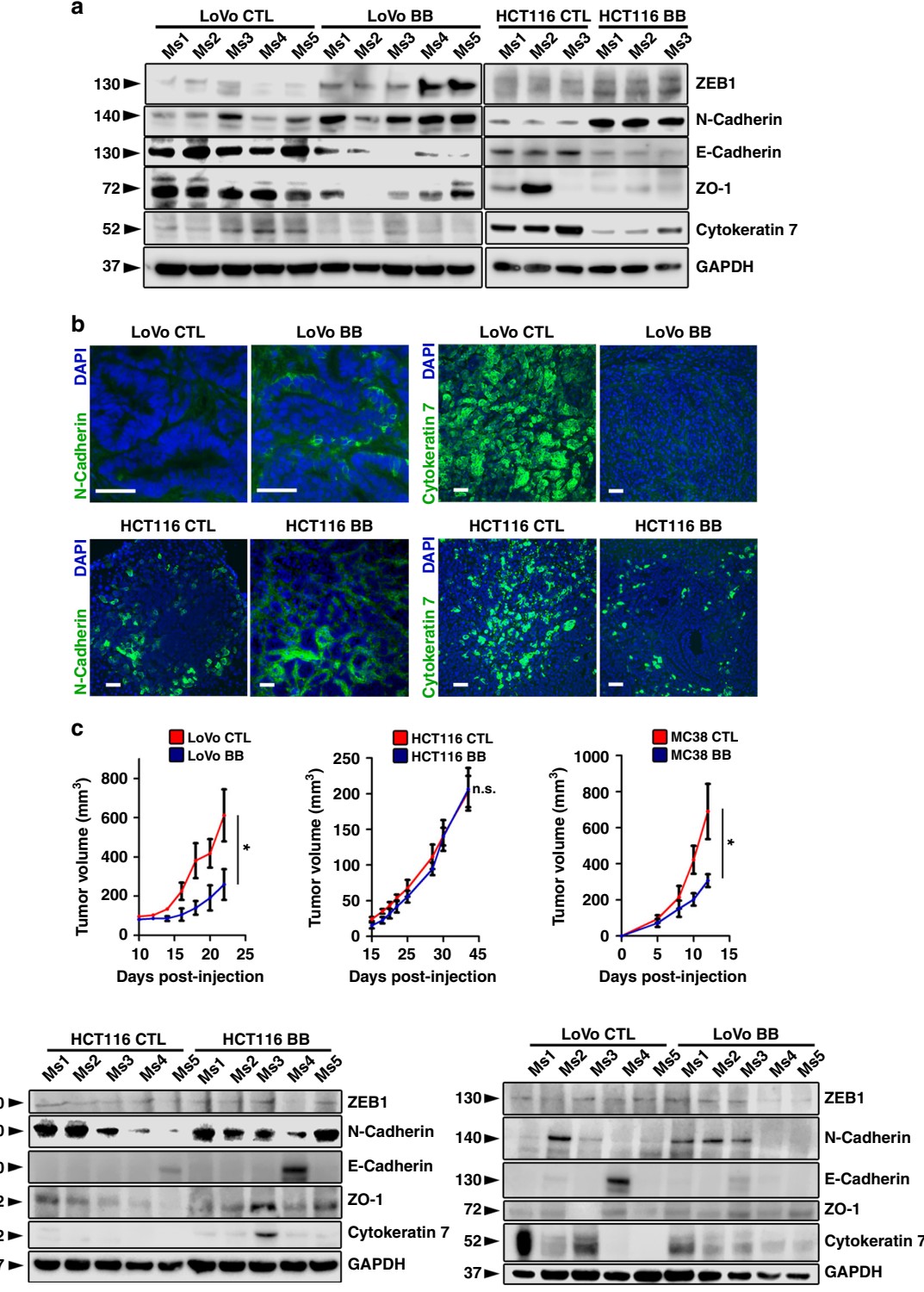

**Fig. 6** Pharmacological inhibition of PADs alters EMT markers in experimental hepatic metastases but not in subcutaneous xenografts. **a** Immunoblotting for the indicated EMT proteins performed on tissue lysates from the experiment in Fig. 5f. GAPDH was used as a loading control. **b** Representative images of staining for indicated proteins in experimental hepatic metastases from Fig. 5f. At least five biological replicates were stained per group. Scale bar = 100 μm. **c** Subcutaneous tumor growth curves of mice injected with indicated cells and treated with vehicle ($n = 5$, 5, and 6 biological replicates) or BB-Cl-amidine ($n = 5$, 5, and 5 biological replicates). Repeated twice. **d** Immunoblotting for the EMT marker proteins as indicated performed on tissue lysates from (**c**). For **c**, error bars indicate s.e.m. (*$P < 0.05$, n.s. = non-significant, two-way ANOVA). Ms – mouse. CTL control, BB BB-Cl-amidine

compared to adjacent colonic mucosa[34]. The release of PAD4 from the cancer cells is clearly complex. Part of this release may be through EVs. This could be analogous to the release of IL-1β or other secreted molecules which also lack a secretory peptide and can

be secreted in EVs, which then through unknown mechanism release their contents into the extracellular space[26,35,36]. The release through EVs is supported by our finding of enrichment of vesicle contents in general within the cancer ECM.

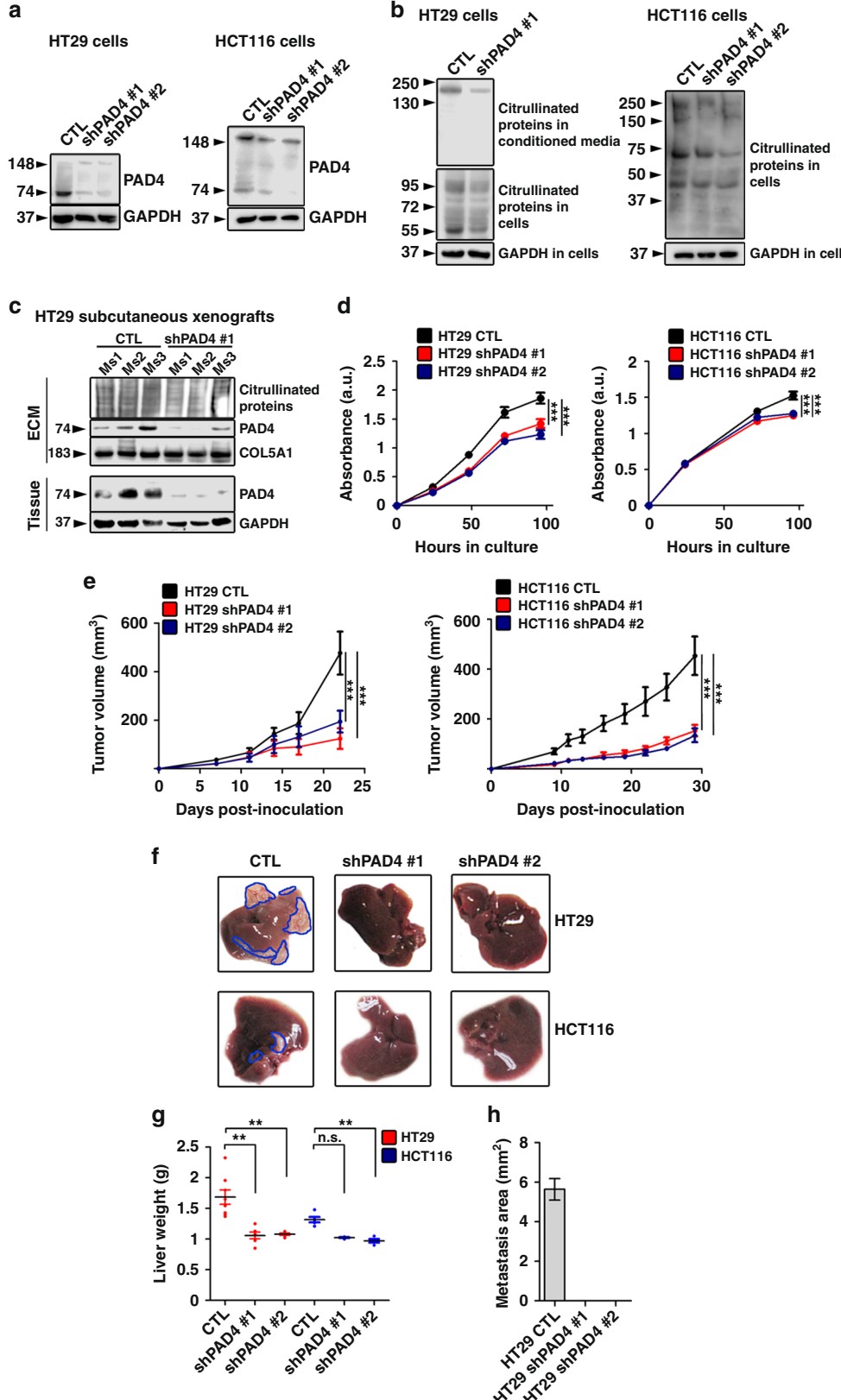

Other molecules we identified in the ECM of metastatic liver also have the capacity to affect EMT. Versican, which was considerably overexpressed in liver metastasis ECM, has been reported to induce MET[37]. TGF-β signaling has been recognized as critical to CRC liver metastasis[38] and we found upregulation of the TGF-β binding proteins LTBP1-3, TSP1, and TSP2, some of which contribute to TGF-β activation[39,40]. TGF-β has been linked to tumor suppression through EMT[42], and citrullination of TGF-β altered its activity[42]. The complex balance between these signals and the function of epithelial-mesenchymal plasticity in cancer is

**Fig. 7** Genetic downregulation of *PAD4* diminishes proliferation and subcutaneous tumor growth of CRC cells, and reduces experimental metastasis in vivo. **a** Immunoblotting for PAD4 in HT29 and HCT116 cells transfected with lentivirus harboring scrambled shRNA (CTL) or shPAD4 (shPAD4 #1 and #2). GAPDH was used as a loading control. **b** Immunoblotting for citrullinated proteins in conditioned media and cell lysates from control and *PAD4*-deficient HT29 and HCT116 cells. GAPDH was used as a loading control. **c** Immunoblotting analysis for PAD4 and citrullinated proteins in tissue lysates and isolated ECM from control (CTL) and *PAD4* knockdown (shPAD#1) subcutaneous xenografts. GAPDH and COL5A1 were used as loading controls for tissue lysates and isolated ECM, respectively. **d** WST-1 proliferation and viability assay performed in cultured wild-type and *PAD4*-deficient HT29 and HCT116 cells ($n = 5$ technical replicates per group). **e** Tumor growth curves of mice injected subcutaneously with control and *PAD4*-knockdown HT29 and HCT116 cells ($n = 7$, 4, and 5 biological replicates for HT29 and 9, 5, and 5, for HCT116, respectively). Repeated twice. **f** Representative images of livers from mice injected intrasplenically with control ($n = 8$ biological replicates) or *PAD4*-deficient HT29 and HCT116 cells ($n = 6$ and 6 biological replicates for shPAD4 #1 and #2) 35 days postinjection. Metastatic nodules are outlined in blue. **g** Liver weights from the experiment in (**f**). **h** Measurement of the area of metastatic deposits in HT29 livers from the experiment in (**f**). For **g**, error bars indicate s.e.m. center values indicate mean (\*\*$P < 0.01$, Kruskal–Wallis test with Dunn's multiple comparison post-test). For **d** and **e**, error bars indicate s.e.m. center values indicate mean (\*\*\*$P < 0.001$, two-way ANOVA). Ms mouse, CTL control, shPAD4 *PAD4* knockdown, ECM extracellular matrix

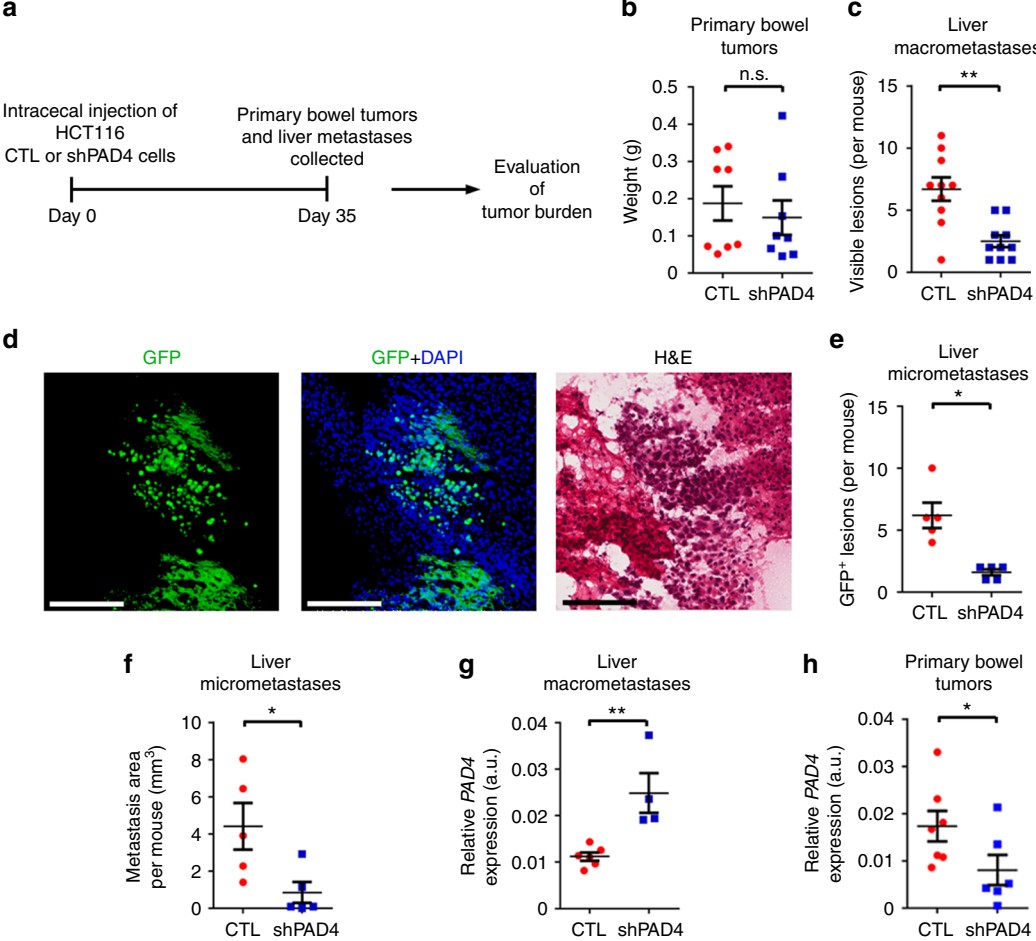

**Fig. 8** Genetic downregulation of *PAD4* in HCT116 cells diminishes spontaneous hepatic metastases in vivo. **a** Experimental pipeline. **b** Weight of excised primary cecal tumors ($n = 8$ biological replicates per group). **c** Number of visible macrometastases ($n = 10$ biological replicates per group). Other organs including lungs showed no evidence of metastasis. **d** Representative image of a GFP$^+$ metastatic lesion counterstained with DAPI (left and middle) and the corresponding H&E image (right). Scale bar $= 100$ μm. **e** Numbers of GFP$^+$ micrometastatic lesions observed ($n = 5$ biological replicates per group). **f** Area of micrometastases evaluated by H&E ($n = 5$ biological replicates per group). **g** Relative (to HPRT) mRNA levels of *PAD4* in visible macrometastases ($n = 6$ and 4 for CTL and shPAD4 groups, respectively). **h** Relative (to HPRT) mRNA levels of *PAD4* in primary cecal lesions ($n = 7$ and 6 for CTL and shPAD4 groups, respectively). Throughout, error bars indicate s.e.m. center values indicate mean (\*$P < 0.05$, \*\*$P < 0.01$, n.s. = non-significant, Mann–Whitney U test). CTL control, shPAD4 *PAD4* knockdown, H&E hematoxylin and eosin

still not completely understood. Genetic manipulation of EMT in murine cancer models leads to increased invasion and migration by metastatic cells[43] while more recent work implicates EMT in promotion of tumor-initiating cells and resistance to chemotherapy[44,45]. After dissemination, MET has been suggested to be necessary to initiate metastatic colony formation[46–48].

Further, MET has been documented in human CRC liver metastases as demonstrated by Hur et al.[49], showing markers for MET in liver metastases compared to their matched primary lesions. This is consistent with our data and the hypothesis that cancer cell secretion of PAD4 leads to MET due to modification of the metastatic liver ECM by citrullination with the result of

facilitating metastatic growth. This hypothesis points to cancer cell plasticity being enforced by an extracellular signal, that of citrullination of ECM components. This novel mechanism could be exploited by cancer cells to secure their growth at distant sites, and interruption of this signaling could generate an unfavorable microenvironment for cancer cell growth. Therefore, PAD4 and citrullination might be a possible target in therapy of liver metastases.

Extensive protein crosslinking and insolubility makes separation of the ECM from cellular contents difficult and proteomic analysis of the tumor ECM is therefore challenging. Here for the first time we demonstrate a method for high-throughput quantitative proteomics of tumor ECM, in which efficacy was independently validated by other methods and could be used to detect protein modification. These methods should be widely applicable to future studies of cancer ECM.

Finally, our group further demonstrated the significance of cancer matrisome in a recent bioinformatics study by implicating a combination of nine core ECM proteins in predicting cancer survival across large cohorts of cancer patients[50].

## Methods

**Human samples.** Human primary CRC and uninvolved colon from the same resection specimens as well as hepatic CRC metastasis tissue and surrounding unaffected liver tissue from resection specimens were obtained from patients from the Oxford Radcliffe Biobank following institutional review and granting of ethical approval (ethics number 09/H0606/5). Informed consent was obtained from all subjects involved.

**Cell lines.** The following cell lines were used in the study: SW480, SW620, DLD-1, HT29, HCT116, LoVo (all human colon cancer), MC38 (mouse colon cancer), OE33 (human esophageal cancer), and HeLa (human cervical cancer). All the cell lines were purchased from American Type Culture Collection (ATCC), except for MC38 and OE33 which were purchased from Kerafast and Sigma, respectively. Early passage SW480, SW620, DLD-1, HT29, HCT116, and OE33 cell lines were cultured in Roswell Park Memorial Institute (RPMI-1640) media (R8758; Sigma). Early passage MC38, and HeLa cell lines were cultured in Dulbecco's Modified Eagle Medium (DMEM) media (D2429; Sigma). All media were supplemented with 10% fetal bovine serum (FBS, 16000-044; Thermo Fisher), and 100 IU/ml penicillin/streptomycin (15140-122; Thermo Fisher). All cells were cultured under sterile conditions and maintained in a 37 °C incubator with 5% $CO_2$, regularly tested for mycoplasma and genetically authenticated by CRUK Cancer Centre Genomics Facility, Leeds, UK. Cell lines were passaged upon reaching ∼80–90% confluency and their morphology was regularly checked to ensure the absence of cross contamination between cell lines or mycoplasma contamination. To determine cell number, the automated cell counter NucleoCounter NC-100 (Chemometec) or manual hemocytometers were used.

**Cell transfection.** For the generation of cell lines stably expressing green fluorescent protein (GFP), cells were infected with a lentivirus containing a gene encoding GFP driven by CMV promoter (CLS-PCG-8; Qiagen). For transfection, $5 \times 10^3$ cancer cells were spun at 3000 r.p.m. After that, cells were kept for 30 min at 4 °C in media supplemented with 8 µg/ml polybrene (H9268; Sigma-Aldrich) and lentivirus particles at various multiplicities of infection (2–8). After spinoculation, cells were plated in 12-well plates and cultured in a 37 °C incubator with 5% $CO_2$. Two days post transfection, lentivirus-containing media was replaced with fresh media containing puromycin (P8833; Sigma-Aldrich) at a pre-determined concentration (depends on a cell line). Cells were FACS-sorted for GFP after reaching confluency. Following sorting, cells were cultured in puromycin-containing media at a pre-determined concentration. For PAD4 knockdown, lentiviral particles were purchased from Origene (TL310630) and transfection was performed in the same way as described above.

For PAD4 overexpression, the pCHD-CMV-MCS-EF1-Hygro vector (System Biosciences) was used with the firefly luciferase gene cloned in the first MCS and human PAD4 cloned in the second MCS. Control cells were transfected with lentivirus harboring a scramble shRNA. Cell transfection using this vector was performed in accordance with manufacturer's instructions.

**BB-Cl-amidine.** The BB-Cl-amidine compound was synthesized and provided by Paul R Thompson, at the Department of Biochemistry and Molecular Pharmacology, University of Massachusetts Medical School, USA. Lyophilized BB-Cl-amidine was diluted in 100% DMSO to a concentration of 40 mg/ml, aliquot and stored at −20 °C before use. For in vivo administration, BB-Cl-amidine was diluted in PBS to reach the desired concentration of 7.5 mg/kg.

**Animal experiments.** All animal experiments were performed according to the UK Animal Scientific Procedures Act 1986, institutional guidelines of the University of Oxford, as well as published guidelines for the welfare and use of animals in cancer research[51]. All experiments involving mice were conducted within the limits of the Project License issued by the Home Office, UK (PPL numbers 30/2841 and 30/3413). Animals were housed in specific pathogen-free facilities with controlled conditions of temperature (24 ± 1 °C) and humidity (55 ± 10%), and had access to water and food (rodent pellets) ad libitum. Throughout the whole time of experiment 12 h light/dark cycles were carefully maintained. Female severe combined immunodeficiency (SCID) mice aged 6–8 weeks were purchased from ENVIGO (Bicester, UK). Female C57BL/6 mice aged 6–8 weeks were purchased from Charles River Laboratories (Kent, UK). Prior to tissue harvest, adult C57Bl/6 mice were humanely culled by means of intraperitoneal injection of 200 µl pentobarbitone, and organs were dissected thereafter. For hepatic metastases model, mice were first anesthetized using vaporized isoflurane. After that, the upper lateral abdominal wall was incised and $1 \times 10^6$ HT29, HCT116, or LoVo cells, or $5 \times 10^5$ MC38 cells in 100 µl PBS were injected into parenchyma of the spleen. The spleen was removed by electrocautery 1 min post-injection to prevent primary tumor growth in the splenic bed. The wound was then closed using non-soluble sutures and autoclips. SCID mice injected with HT29, HCT116, or LoVo cells were sacrificed 35 ± 2 days following surgery. C57BL/6 mice injected with MC38 cells were sacrificed 13 ± 2 days following surgery. For intraceal injection, mice were first anesthetized using vaporized isoflurane. After that, a transverse upper abdominal wall incision was made, the cecum was exteriorized, and $1 \times 10^6$ wild-type or shPAD4 HCT116 cells in 50 µl PBS were slowly injected between the mucosa and the muscularis layers of the cecal wall. After injection, the cecum was returned to the abdominal cavity and the wound was closed with surgical sutures and autoclips. For in vivo PAD4 inhibition experiments, randomized SCID or C57BL/6 mice were administered with 7.5 mg/kg BB-Cl-amidine or 3–5% DMSO in 100 µl PBS intraperitoneally every 48 h starting from the day 1 after the intrasplenic or subcutaneous injection. For neutrophil depletion experiments, 12.5 µg/kg of anti-Ly6G (clone 1A8; BD Biosciences, Oxford, UK) was injected intrasplenically every 24 h. An isotype control antibody immunoglobulin [Ig] G2a was injected as a control. Mice were randomized before the start of the treatment, using a random number generator. No animals were excluded.

**Liver decellularization optimization.** Livers were obtained from mice after sacrifice. For whole liver decellularization, different detergents were tested. Dissected mouse livers were washed in phosphate-buffered saline (PBS) and placed in Petri dishes with 4 different solutions: (1) 1% sodium dodecyl sulfate (SDS) and 0.01% ammonium hydroxide ($NH_4OH$) in dd$H_2O$, (2) 2% TritonX-100 and 0.1% $NH_4OH$ in dd$H_2O$, (3) 1% TritonX-100 diluted in PBS, and (4) 1% SDS diluted in PBS. The dishes were placed on a rotary shaker for 72 h; solutions were changed every 6 h during the day, and left at −4 °C overnight. Decellularization was assessed visually every 12 h, and images were taken using a digital camera. As a result, 1% SDS and 0.01% $NH_4OH$ in dd$H_2O$ was considered the best buffer for decellularization.

**ECM preparation.** Human hepatic metastasis tissues and adjacent unaffected liver tissues were decellularized in a buffer containing 0.1% SDS and 0.01% $NH_4OH$ in dd$H_2O$ for 72 h, cut into 100 mg pieces (wet weight), aliquot into 200 µl of ice cold buffer C (N-2-hydroxyethylpiperazine-N-2-ethane sulfonic acid (HEPES) pH 7.9, $MgCl_2$, KCl, EDTA, sucrose, glycerol, sodium orthovanadate, protease inhibitor cocktail) of the CNMCS Compartmental Protein Extraction Kit (K3013010; BioChain Institute), flash frozen in liquid nitrogen and kept at −20 °C until enrichment.

**ECM enrichment.** Frozen ECM samples were thawed, then homogenized using the Tissue-Tearor tissue homogenizer (BioSpec Products) at speed 4 or 5 for 25 s on ice. Homogenization was repeated two to three times; the probe was cleaned with 100% ethanol and PBS between samples. The samples were further processed on the high intensity VCX-130 ultrasonic processor (Sonics&Materials) at 4 °C with the following settings: 100% amplitude, 2 s on/2 s off, 20 s in total (including offs). The samples were then deglycosylated with 3000–4000 units of PNGase F (P0704; New England BioLabs) and contaminating nucleic acids were removed by adding 1 µl Benzonase® nuclease (E1014; Sigma) for 1 h at 4 °C. The samples were then spun down at 18,000 × $g$ for 20 min. The supernatant was discarded and the pellet was washed in 400 µl of ice cold buffer W (HEPES (pH7.9), $MgCl_2$, KCl, EDTA, sucrose, glycerol, and sodium orthovanadate, hereinafter, concentration of each component is not specified by a supplier) of the CNMCS Compartmental Protein Extraction Kit at 4 °C for 5 min. The washed protein extract was spun at 18,000 × $g$ for 20 min. The supernatant was discarded, and the pellet was resuspended in 150 µl of ice cold buffer N (HEPES (pH7.9), $MgCl_2$, NaCl, EDTA, glycerol, sodium orthovanadate) and incubated at 4 °C for 20 min to solubilize nuclear proteins. Protein extract was spun at 18,000 × $g$ for 20 min. The supernatant was discarded, and the pellet was resuspended in 150 µl of ice cold buffer M (HEPES (pH7.9), $MgCl_2$, KCl, EDTA, sucrose, glycerol, sodium deoxycholate, NP-40, sodium orthovanadate) for membrane proteins solubilization. The extracts were then spun down at 18,000 × $g$ for 20 min, and the supernatant was discarded. The remaining

pellet was resuspended in 150 μl of pre-warmed buffer CS (Pipes (pH6.8), MgCl₂, NaCl, EDTA, sucrose, SDS, and sodium orthovanadate) and incubated at RT for 20 min to solubilize cytoskeletal proteins. Protein extract was spun at $18,000 \times g$ for 20 min. Finally, the supernatant was withdrawn, and the pellet was resuspended in 150 μl of buffer C, incubated at 4 °C for 5 min and then spun for 20 min at $15,000 \times g$ at 4 °C. The remaining insoluble pellet consisting of ECM proteins was then flash frozen in liquid nitrogen and kept at −20 °C.

**Biochemical processing of ECM-enriched proteins**. ECM-enriched samples were solubilized in 8M urea (9U5378; Sigma), 100 mM ammonium bicarbonate (09830; Sigma), 10 mM dithiothreitol (DTT, 43817; Sigma), pH 7.8, and incubated at 37 °C for 30 min. Alkylating agent iodoacetamide (I1149; Sigma) was added to a final concentration of 25 mM and the samples were incubated for 30 min at RT. Protein samples were then precipitated via the routine methanol/chloroform method, followed by resuspension in 50 μl 8M urea and vortexing. Finally, the urea concentration was reduced to a final concentration of <1M by diluting the reaction mixture with ddH₂O. At this point, to determine the concentration of protein for the proteomics, we used the DC™ (detergent compatible) protein assay according to instructions (5000111; Bio-Rad). ECM-enriched pellets were digested overnight using trypsin (V5111; Promega), at a ratio of 1:50 enzyme:substrate. The samples were constantly vortexed on a shaker at 37 °C. A second aliquot of trypsin was added at a ratio of 1:100 enzyme:substrate, and samples were incubated for additional 4–6 h at 37 °C with shaking.

**Mass spectrometry**. The samples were processed with nano–liquid chromatography tandem mass spectrometry (nano–LC-MS/MS), utilizing the Acquity LC instrument (C18 column with a 75 μm × 250 mm, 1.7 μm particle size; Nanoacquity Waters) coupled to a Thermo LTQ Orbitrap Elite mass spectrometer (resolution of 120,000 at 400 m/z, Top 20, collision-induced dissociation), using a gradient of 1–35% acetonitrile for 60 min at a flow rate of 250 nl/min. Peak lists containing MS/MS spectra were generated using MSConvert (Proteome Wizard). After that, these lists were searched using Mascot version 2.3 (http://www.matrixscience.com), against the Swiss-Prot protein database containing mouse (16,642 entries as of September 2012) or human (20,306 entries as of June 2014) sequences, with tryptic restriction and mass deviations of 10 parts per million/0.5 daltons in the respective MS modes. Oxidation of methionine, deamidation of asparagine and glutamine, and other known collagen and proteoglycan modifications were set as variable modifications. Peptide false discovery rate was adjusted to 1%. For label-free quantification of differentially expressed proteins, normalized abundance of each protein was determined by measuring the peak area intensity utilizing the Progenesis QI software (Nonlinear Dynamics). Briefly, protein abundance was calculated from the sum of all unique peptide ion abundances for a specific protein on each run. Normalization of abundance was performed to allow comparisons across different sample runs by the software. Proteins identified by more than one peptide were retained. The normalized peptide intensities for each sample were used to calculate fold change ratios for proteins between sample groups.

**Protein categorization**. ECM proteins were split into five major groups: proteoglycans, glycoproteins, collagens, ECM regulators, ECM-affiliated proteins and secreted factors according to the categorization proposed by Naba et al. (2012). Gene Ontology term enrichment analysis for cellular localization was performed using the PANTHER bioinformatics resource (http://pantherdb.org/).

**LC-MS/MS detection of citrulline residues**. LC-MS/MS data were analyzed in PEAKS® software (V.6; Bioinformatics Solutions, Waterloo, ON, Canada) using the posttranslational modification search option at mass tolerances of 10 ppm for MS1 and 0.5 Da for MS2 at 1% false discovery rate (FDR). Citrullinated peptides were manually inspected when the citrullinated residue produced a missed cleavage. To exclude false positive assignment of citrullination, we confirmed correct precursor and fragment mass assignments (monoisotopic mass instead of ¹³C isotopic peak) and also excluded deamidation, which exhibits the same mass shift.

**Antibodies**. The list of antibodies used in the study is provided in the Supplementary Data 4.

**In vitro citrullination studies**. For all experiments involved in vitro citrullination, recombinant collagen I (ab6308; Abcam) was incubated at a concentration of 40 μg/ml with 200 ng active human PAD4 (ab196393; Abcam) in the presence or absence of 1 mM or 0.05 mM BB-Cl-amidine for 24 h at 37 °C in a buffer of 20 mM HEPES (pH 8.8), 10 mM CaCl₂, 1 mM dithiothreitol (DTT), 1 mM ethylenediaminetetraacetic acid (EDTA), and 0.3 M NaCl. The concentrations of PAD4 and collagen I were used to obtain the enzyme-to-substrate ratio of 1:10. Samples were analyzed by LC-MS/MS for the presence of citrulline residues. For the coating experiments, collagen density used was 10 μg/cm². The mixture of collagen I and PAD4 (and BB-Cl-amidine if needed) was prepared and immediately added into wells (96-well, 24-well, or 6-well) and then incubated for 24 h at 37 °C in cell culture incubator to prevent drying. Before plating the cells, the wells were dried in a 37 °C incubator for 1–2 h and carefully washed with PBS three times.

**Immunofluorescence and H&E staining**. Liver metastasis tissues, adjacent unaffected liver tissues, and murine- or human-derived decellularized matrices were embedded in optimal cutting temperature (OCT) compound (25608-930; Tissue-Tek), flash frozen and kept at −80 °C until cutting. Tissue sections were prepared using the OTF 5000 cryostat (Bright instrument), and 12 μm slices were mounted on Superfrost Plus microscope slides (4951PLUS4; Thermo Fisher). For immunofluorescence, the sections were brought to RT, fixed in ice-cold acetone (or 4% paraformaldehyde or methanol depending on antibody), rehydrated, blocked with 20% goat serum, and incubated overnight with the primary antibodies. The following day, the slides were washed in PBS before being incubated with secondary antibodies for 1 h at RT. The sections were washed and mounted using the Pro-Long® Diamond Antifade Mountant with DAPI (P36962; Thermo Fisher). Immunofluorescence was visualized utilizing an inverted epifluorescence microscope (DM IRBE, Leica Microsystems) with a digital camera. All images were acquired with either a ×10 or a ×20 objective. The excitation source was CoolED pE-300 ultra. The images were processed using either ImageJ (National Institutes of Health, Bethesda, MD) or Adobe Photoshop CS4 (Adobe Systems). For H&E staining, slides were fixed in ice-cold acetone for 10 min, then immersed in filtered Harris modified Hematoxylin solution (HHS16; Sigma) for 1 min, then rinsed with tap water. The sections were immersed in 1% aqueous Eosin Y solution (HT110216; Sigma) for 1–2 min and rinsed with tap water. The slides were dehydrated in ascending alcohol solutions (50%, 70%, 80%, 95% × 2, and 100% × 2) and cleared with xylene (214736; Sigma) two times for 5 min each. Finally, coverslips were mounted onto a labeled glass slide with Permount Mounting Medium (SP15-100; Thermo Fisher). For quantification of liver metastatic areas, percent of metastatic area in H&E-stained sections of comparable size was quantified using the ImageJ software.

**Protein extraction from cells and tissues**. For protein extraction from cells, cell pellets were lysed with RIPA lysis buffer (89900; Thermo Fisher), containing a Complete Protease Inhibitor Cocktail (04693116001; Roche) or Protease/Phosphatase Inhibitor Cocktail (5872; CST) at a ratio of 5:1 or 1:100, respectively for 30 min at 4 °C on a rotary shaker. For protein extraction from intact or decellularized tissues, 0.5 g of freshly dissected tissue was disrupted at 4 °C using an electric Tissue-Tearor homogenizer, passed through a 70 μm cell strainer and spun down. Cell pellets were resuspended in RIPA lysis buffer containing a protease inhibitor at a ratio of 5:1 for 2 h at 4 °C on a rotary shaker. The extracts were briefly sonicated and then spun down at $12,000 \times g$ at 4 °C for 10 min to remove insoluble material. Protein concentration was determined using the BCA protein assay kit (23225; Thermo Fisher). All protein suspensions were flash frozen in liquid nitrogen and kept at −20 °C until use.

**Extracellular vesicles (EV)**. Cells were plated at high density ($10 \times 10^6$) and cultured for 48 h in DMEM supplemented with 0.1% EV-free media (depleted of EVs by overnight centrifugation at $100,000 \times g$). Conditioned media was collected and sequentially centrifuged at $2000 \times g$ for 30 min, then $10,000 \times g$ for 40 min, then $100,000 \times g$ for 18 h using Optima™ XPN-80 Ultracentrifuge (Beckman Coulter) and SW 32 Ti Rotor. At this point samples were removed for ELISA for PAD4. Resulting pellets were washed with PBS and centrifuged at $100,000 \times g$ for 1.5 h using Optima™ MAX-XP Ultracentrifuge (Beckman Coulter) and TLA-55 rotor. All the centrifugations were performed at 4 °C. Resulting pellets were processed for transmission electron microscopy analysis, immunoblotting, or nanoparticle tracking analysis. Protein concentration was normalized against number of plated cells or measured using microBCA assay (23235; Thermo Fisher). Uncropped immunoblotting images are presented in Supplementary Fig. 13.

**GW4869**. GW4869 was purchased from Sigma (D1692). Cancer cells were plated at high density ($10 \times 10^6$) and serum starved for 8 h. Media was changed for Opti-MEM™ (31985070; Thermo Fisher) and 20 μM GW4869 was added to cells at time points 0, 24, and 48 h. The conditioned medium was collected at 72 h and EVs were isolated as described above. HT29 cells were unaffected in the WST-1 assay by GW4869, but LoVo and HCT116 had reduced signal as a result, and EVs were not collected after treatment for these cell lines (Supplementary Fig. 8C).

**SDS-PAGE and gel silver staining**. For the experiments shown in Supplementary Fig. 1 and Supplementary Fig. 2, total protein (5 μg) obtained from intact tissue, decellularized tissue, or decellularized biochemically enriched tissue, were mixed with LDS sample buffer (NP0008, Life Technologies) at a ratio 4:1, and then loaded on 1.0 mm 3–8% Tris-Acetate protein gradient gel (EA0375, Life Technologies). Proteins were separated at 100 V for 2 h. Silver staining of gels was performed using the SilverQuest staining kit (LC6070; Life Technologies) according to the manufacturer's instructions. Images of the stained gels were acquired with the PhotoDoc-It UVP Imaging System (Cole-Parmer, UK).

**Immunoblot analysis**. For immunoblotting, an equal amount of protein lysates (8–60 μg, depending on the primary antibody used) were mixed with LDS sample buffer at a ratio 4:1, and loaded on either 3–8% Tris-Acetate or 4–12% Bis-Tris precast gels (NP0321; Life Technologies) for high and low molecular weight proteins, respectively. Proteins were separated at 100 V for 1.5–3 h and transferred at

30 V for 1–1.5 h 4 °C using the polyvinylidene fluoride membrane (IPVH00010; Millipore). Membranes were then blocked in either 5% bovine serum albumin (A2153; Sigma) or 5% skimmed milk diluted in TBS supplemented with 0.05% Tween-20 (P7949; Sigma) (TBST). Blots were incubated overnight at 4 °C with the primary antibodies at dilutions ranging from 1:100 to 1:1000. The membranes were washed three to five times for 10 min each in TBST, then horseradish peroxidase conjugated secondary antibodies (all Santa-Cruz) were added at dilution 1:500 for 1 h at RT. The membranes were then washed three times for 10 min each in TBST. Proteins were visualized with enhanced chemiluminescence by using the Amersham ECL detection reagent (RPN2106; GE Healthcare) or the Odyssey® imaging system (LI-COR) or an X-ray detector. Densitometric analysis of band intensity was performed using the ImageJ software or the Image Studio™ Software (LI-COR). Uncropped blots are provided in the Supplementary Information.

**Citrullination immunoblot analysis**. For determination of citrulline residues in liver metastases tissues and adjacent unaffected liver tissues, a modified anti-citrulline detection kit (17-347; Merck Millipore) was used according to manufacturer's instructions. The assay principle is based on immunoblot technique using the anti-citrulline antibody. To detect citrulline residues, the polyvinylidene fluoride membrane with transferred proteins was exposed overnight to 2,3-butanedione monoxime and antipyrine in a strong acid solution ($H_2SO_4$) to form ureido group adducts to citrulline residues. This ensures specific detection of citrulline containing proteins disregarding any other amino acid sequences.

**ELISA**. For determination of citrulline residues in liver metastases tissues and adjacent unaffected livers, an anti-citrulline human ELISA kit (MBS701429, MyBioSource) was used according to manufacturer's instructions. For determination of PAD4 levels in liver metastases tissues and adjacent unaffected livers, an anti-PAD4 human ELISA kit (LS-F7962, LifeSpan BioSciences) was used according to manufacturer's instructions.

**FACS and cell isolation**. Dissected mouse livers were minced with a scalpel on ice and incubated at 37 °C in RPMI-1640 media containing 0.05% collagenase/dispase (11097113001; Roche) and 0.01% trypsin inhibitor (T7659; Sigma). Cell suspension was filtered through a 70 μm cell strainer into PBS and residual red blood cells lysed using red cell lysis buffer (11814389001; Roche). Cell suspensions at a concentration of $1 \times 10^7$ cells/ml were the incubated for 45 min at 4 °C in the presence of anti-CD11b PE-Cy7 (25-0112-82, eBioscience), anti-Gr1 PE (12-5931-82; eBioscience), anti-CD45 PE (12-0451-82; eBioscience), anti-CXCR2-APC (149305; Biolegend), and Mouse BD Fc Block (553141, BD Bioscience). FACS analysis was performed using a FACSCalibur flow cytometer (BD Biosciences) and analyzed with FlowJo software version 7.2.5 (Tree Star, Ashland, OR).

**Viability and proliferation assay**. In total, $5 \times 10^3$ cells were seeded in 96-well plates. Five to seven technical replicates per condition were used. After 24 h of culture, media was replaced with fresh media containing 10% WST-1 (ab155902; Abcam) and cells were left in the incubator for 1 h. Absorbance was read at 450 nm using the plate reader. For experiments in Supplementary Fig. 9, we used a very similar Cell Cytotoxicity Assay Kit (ab112118; Abcam). Here, after 24 h of culture, media was replaced with fresh media containing 20% ab112118 and cells were left in the incubator for 3 h. Absorbance was read at 570 and 605 nm using the plate reader, and their ratio was used to evaluate viability/proliferation.

**Adhesion assays**. In total, $5 \times 10^4$ GFP⁺ cells were seeded in triplicates in 6-well plates pre-coated overnight (24 h) with collagen I, collagen I + PAD4, or collagen I +PAD4+BB-Cl-amidine (for details see "In vitro citrullination studies"). At 1, 2, and 3 h time points, the plates were imaged using an inverted epifluorescence microscope at ×10 objective (DM IRBE, Leica Microsystems, at least five fields of view per condition) and the number of adherent cells was manually counted.

**Time-lapse microscopy**. In total, $1 \times 10^3$ GFP⁺ cells were seeded in 96-well plates pre-coated overnight (24 h) with collagen I, collagen I + PAD4, or collagen I +PAD4+BB-Cl-amidine (for details see "In vitro citrullination studies"). Wells were washed with PBS three times to remove PAD4 or BB-Cl-amidine before cell plating. Sixteen technical replicates were used per condition. After adherence, cells were imaged for 20 h using the epiflourescent time-lapse microscope Nikon Ti-E at ×10 objective. Images were taken every 45 min. After the imaging session, individual cells were tracked using the IMARIS software (Bitplane). Median cell track distance and speed were analyzed.

**mRNA extraction and real-time quantitative PCR**. Total RNA was isolated from FACS-sorted cells using RNeasy kit (74104; Qiagen) or TRIzol reagent (15596026, Thermo Fisher), and then TURBO™ DNase kit was used to deplete DNA (AM2238; Thermo Fisher). RNA concentration was measured using NanoDrop microvolume spectrophotometer (Thermo Fisher). Samples were reverse transcribed using Tetro cDNA synthesis kit (BIO-65042; Bioline) according to manufacturer's instructions. Resulting complementary DNA (cDNA) was analyzed using SYBR Green technology in the Mx3000P instrument (Stratagene). Gene expression was compared by the way of the ΔΔCt method after normalization to hypoxanthine guanine phosphoribosyl transferase (HPRT) expression. The following real-time PCR primers were used: human PAD4 forward (5′-TTGCAATCAACTGGAGCAGG-3′) and reverse (5′-AGAGGGCTGAGGCCACCT-3′), mouse PAD4 forward (5′-TTACCACGGAGTTCCACACC-3′) and reverse (5′-CTGCCAGCCATAGTTAAGC-3′), human HPRT forward (5′-ATAGGACTCCAGATGTTTCC-3′) and reverse (5′-ATAAGCCAGACTTTGTTGG-3′), mouse HPRT forward (5′-GCAGTACAGCCCCAAAATGG-3′) and reverse (5′-AACAAAGTCTGGCCTGTATCCAA-3′).

For all other genes, pre-validated and optimized primers were ordered from Sigma (KiCqStart®). For analysis of EMT gene expression in MC38 cells (Fig. 4f), the commercially available gene array was used (PAHS-090Z; Sabiosciences) and performed in accordance manufacturer's instructions.

**Conditioned media experiments**. Cell conditioned media was concentrated using centrifugal filter tubes (Merck). For Fig. 3e, original media samples were first concentrated, then split in two parts and loaded into two parallel gels. One gel was blotted for PAD4, and another one was Coomassie stained (B7920; Sigma).

**Statistical analysis**. Data were analyzed and statistics were performed in Graph Pad Prism version 5.03. Data are presented as mean ± s.e.m., or median ± interquartile range, or box and whiskers and median, where whiskers indicate range. Values with $P < 0.05$ were considered statistically significant. FDR was used to validate peptide and protein hits obtained during the quantitative LC-MS/MS analysis. Two-way ANOVA was used to identify the difference between groups in the quantitative proteomics analysis, subcutaneous tumor growth curves, and viability/proliferation assays. For analysis of two groups with unpaired samples, Mann–Whitney U test was used. For analysis of two groups with paired samples, Wilcoxon signed rank test was used. In Fig. 2b, values in both groups were normally distributed (Kolmogorov–Smirnov test: $P > 0.1$), and therefore paired Student's $t$ test was used. For analysis of three or more groups, Kruskal–Wallis test with Dunn's multiple comparison post-test was used. Statistical methods were not used to predetermine necessary sample size, but sample sizes were based on review of similar experiments in literature and previously published results such that appropriate statistical tests could yield significant results. Experiments were not performed in a blinded fashion. The following symbols represent significant differences throughout the figures: *$P = 0.01$ to $0.05$, **$P = 0.001$ to $0.01$, ***$P < 0.001$.

## Data availability
The mass spectrometry proteomics data have been deposited to the ProteomeXchange Consortium via the PRIDE partner repository with the dataset identifier PXD010252 and 10.6019/PXD010252. Use the following link to access the dataset: http://www.ebi.ac.uk/pride/archive/projects/PXD010252

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

## Acknowledgements

The work has been supported by CRUK funding to the Oxford Institute for Radiation Oncology (C5255/A15935), CRUK/EPSRC Oxford Cancer Imaging Centre (C5255/A16466). The authors are thankful to Graham Brown for help with microscopy; Camilla Gomes for help with cryostat work; Sally Hill for help with animal work; and Errin Johnson from the Sir William Dunn School of Pathology (University of Oxford) for help with electron microscopy.

## Author contributions

A.E.Y. and R.J.M. conceived and A.E.Y., R.J.M., and S.Y.L. designed the study. A.E.Y. conducted most of the experiments, A.N.G.-W., B.M., M.L.T. and K.J. performed experiments and interpreted data. A.M. synthesized BB-Cl-amidine. R.K. and R.F. performed the MS/MS analysis, A.N.G.-W., P.J.V., P.R.T., E.O., B.M.K. and S.Y.L. critically analyzed data. A.E.Y., A.N.G.-W. and R.J.M. wrote the manuscript.

## Additional information

**Competing interests:** The authors declare no competing interests.

