## [Peer Review File · Nature Communications]

Reviewers' comments:

Reviewer #1 (Remarks to the Author):

This Ms describes some potentially interesting results. The authors report citrullination of ECM proteins in CRC metastases to liver and explore the possible relevance of this interesting observation for tumor progression and metastasis. Unfortunately, the experiments presented fall far short of clearly indicating any causal consequences of ECM citrullination on metastasis as claimed.

Citrullination of ECM proteins has been studied extensively in inflammatory disorders such as rheumatoid arthritis where the post-translational modifications can both affect cellular interactions with the ECM as well as elicit antibodies leading to autoimmunity. Citrullination of histones is also involved in the induction and release of neutrophil extracellular traps (NETs) Elevated citrullination has been previously reported in cancer.

This Ms concentrates on liver metastases from colorectal cancer (CRC). It is reported that MS analyses of ECM-enriched fractions from CRC liver metastases show elevated citrullination of many ECM proteins and also of other proteins suggested by the authors to originate from exosomes. The arginine deiminase, PAD4, which catalyzes citrullination was also shown to be elevated in CRC liver mets relative to normal liver (Figure 1D). These experiments are convincing so far as they go.

Figure 1 (also Figs S1-S5) all show higher levels of citrullination in liver mets as compared with normal liver but do not contain any comparisons between normal colon and CRC mets, i.e., this is not the relevant comparison. We do not know whether or not citrullination is altered in association with metastasis nor whether this is a difference between CRC and liver or even between normal colon and liver. The subsequent analyses in Figure 2E and F do show elevated citrullination and PAD4 levels in CRC metastases versus primary CRC or normal colon but these analyses are only of overall citrullination, without any analyses of ECM. Furthermore, there are zero details about the sources of these colon and CRC samples. Several CRC cell lines were also reported to show elevated levels of PAD4 and to secrete it into their culture medium (Figure 3). PAD4 is reported here to be associated with exosomes isolated from the culture medium (Fig. 3F-I), although there was no determination of the enrichment on isolation of the particles designated as exosomes and the identification of the particles as exosomes is limited. PAD4 was also shown to be associated with ECM in the CRC liver mets (Fig 3J).

The effects of citrullination of collagen on cell behaviour were then monitored; citrullination enhanced adhesion and reduced migration (Figure 4), although the effects on migration were marginal. Some rather limited assays for integrin-mediated signal transduction events and for mRNA levels of markers for epithelial or mesenchymal state of the cells were then conducted (Fig. 4 E-G) and interpreted as indicating that citrullination predisposed to an epithelial state. These analyses are insufficient to document the authors' conclusion about an MET.

Mice injected with CRC cells were next treated or not with an inhibitor of PAD4 (BB-CI-amidine). The results were rather variable but in general showed reduced citrullination, although ECM proteins were not explicitly assayed. There was a reduction in metastatic burden on treatment with the PAD4 inhibitor and variable indications of reduced mesenchymal markers and increased epithelial markers but there is no evidence that this had anything to do with ECM proteins and shRNA knockdown of PAD4 did not affect EMT markers, although it did reduce metastases in two types of metastasis assays. However, the knockdown of PAD4 markedly reduced growth of subcutaneous primary tumors. This does not really fit well with the earlier claim that there is no change in citrullination in primary CRC.

Again, these experiments need much more thorough analyses to be convincing – there are interesting hints of connections between citrullination, MET and metastasis but also inconsistencies

that need to be followed up and explained.

Furthermore, there needs to be a better connection made between the multiple different strands of the paper; citrullination of ECM, exosomes, in vitro effects of citrullination of collagen, the MET hypothesis and the in vivo results – as discussed, the latter are not very reproducible or consistent. There are interesting preliminary results here but they need to be followed up in much more detail before publication could be recommended.

Reviewer #2 (Remarks to the Author):

Overall, Yuzhalin et al show a novel role for ECM protein citrullination influences liver metastatic growth in colorectal cancer. This is a novel, and well controlled study. The authors should be congratulated on a comprehensive piece of work.

Reviewer #3 (Remarks to the Author):

Liver metastatic growth in colorectal cancer depends on modification on the extracellular matrix by citrullination

By A.E Yuzahlin et al.

In this manuscript the Authors have analysed the extracellular matrix of the colon cancer, healthy colon, liver metastasis and healthy liver tissue by combining human and samples from mouse models and highly demanding technique of tissue decellularization. Their data indicate that extracellular matrix (ECM) of liver metastasis has increased levels of citrullinated proteins compared to all other tissues analysed. This posttranslational modification of proteins has been previously detected in autoimmune disease but the data in cancer is rather scarce thus underlying the novelty of this study. In vitro data suggest that the molecular mechanism that is controlled by this posttranslational modification of proteins may be the mesenchymal to epithelial transition (MET). In addition, in vivo data using PAD (peptidylarginine deiminase) inhibitor and three different colon cancer cell lines show similar results. However, there are some contradictory data when only PAD4 is inhibited that should be addressed before publishing as Authors state that PAD4 dependent citrullination of ECM drives liver metastasis formation. The manuscript is interesting, well written, with well performed experiments and provides novel data of potential interest.

Major points:

1. The use of PAD inhibitor in vivo produced the reduction of protein citrullination and reduction of metastasis formation using 3 different cell lines (LoVo, HCT116 and MC38). In addition, the Authors show the increase in expression of mesenchymal markers in metastatic lesions from mice treated with PAD inhibitor therefore implying that reduction of citrullination is producing the reversion of MET. However, when the expression of only PAD4 is downregulated using shRNAs, although there is a reduction in both the levels of extracellular and intracellular PAD4 as well as in levels of intracellular and extracellular citrullinated proteins no effect on the levels of expression of N-Cadherin and E-cadherin was observed. And still these cells were not able to form liver metastasis using two different cell lines. Therefore these results may imply that:

a) The effect seen on MET upon pharmacological inhibition of PAD is due to inhibition of other members of PAD family

b) PAD4 does not control liver metastasis formation through induction of MET

For that reason Authors are suggested to either provide more evidence on PAD4 regulating MET or

to separate citrullination effect on MET in liver metastasis formation from PAD4 mediated liver metastasis formation.

2. Does overexpression of PAD4 affects the expression of epithelial and/or mesenchymal markers?
3. HCT116 cells when injected intraceum, beside primary tumor could develop both liver and lung metastasis. Is the formation of lung metastasis affected upon PAD4 downregulation in HCT116 cells?

Minor points:

1. In the figure 4A the presented data is according to figure legend from 2 independent experiments. However, error bars were not provided. Please provide this data. In the same figure the effect of PAD inhibitor BB-CI-amidine on the amount of citrullinated peptides is bigger when 20 times lower dose of inhibitor is used (see data on RhColl +PAD4 +1mM BB vs RhColl +PAD4 +0.05mM BB). How the Authors explain this?
2. Please provide better resolution images for the Figures 5C and 6D
3. The Authors claim that GW4869 inhibitor was not tested on HCT116 and LoVo cells due to toxicity. However, the data presented in the figure S6E do not show the effects of toxicity rather the reduction of proliferation. Please comment on this data.
4. Please provide the details in the Figure legend for all the blots presented in the figure S7B.
5. Please provide the explanation why in some blots of citrullinated proteins almost all molecular weights (e.g. S10B) of proteins are presented whereas in some blots only low molecular weight proteins are shown (e.g. Figures 5A and B)? Is BB inhibitor affecting citrullination of only low molecular weight proteins?

Reviewer #1

This Ms describes some potentially interesting results. The authors report citrullination of ECM proteins in CRC metastases to liver and explore the possible relevance of this interesting observation for tumor progression and metastasis. Unfortunately, the experiments presented fall far short of clearly indicating any causal consequences of ECM citrullination on metastasis as claimed.

We agree that we have not proven a causal consequence and we did not intend to leave the reviewers with the impression based on the text that we had done so. We demonstrated that downregulation of PAD4 or pharmacological inhibition of PAD activity greatly reduced experimental liver metastases in several models. And that colorectal cancer cells, plated on collagen type I citrullinated by PAD4 are altered in attachment, migration and reduced mesenchymal marker expression, the latter also varying in vivo based on PAD activity in liver metastases. As a result our data is consistent with the hypothesis that tumor cell derived PAD4 citrullinates the ECM resulting in reduced mesenchymal markers in liver metastases and reduced liver metastatic growth. We have altered the title and rewritten the text throughout to explicitly indicate that this is a hypothesis resulting from the work (see at the end of the file in yellow). In general the text has been extensively rewritten throughout to make these points clear. The paragraphs indicated at the end of the file are examples of our rewritten text to clearly state our hypothesis and to distinguish the hypothesis from the experimental results.

-Citrullination of ECM proteins has been studied extensively in inflammatory disorders such as rheumatoid arthritis where the post-translational modifications can both affect cellular interactions with the ECM as well as elicit antibodies leading to autoimmunity. Citrullination of histones is also involved in the induction and release of neutrophil extracellular traps (NETs)

Elevated citrullination has been previously reported in cancer.

This Ms concentrates on liver metastases from colorectal cancer (CRC). It is reported that MS analyses of ECM-enriched fractions from CRC liver metastases show elevated citrullination of many ECM proteins and also of other proteins suggested by the authors to originate from exosomes. The arginine deiminase, PAD4, which catalyzes citrullination was also shown to be elevated in CRC liver mets relative to normal liver (Figure 1D). These experiments are convincing so far as they go.-

Figure 1 (also Figs S1-S5) all show higher levels of citrullination in liver mets as compared with normal liver but do not contain any comparisons between normal colon and CRC mets, i.e., this is not the relevant comparison. We do not know whether or not citrullination is altered in association with metastasis nor whether this is a difference between CRC and liver or even between normal colon and liver.

The subsequent analyses in Figure 2E and F do show elevated citrullination and PAD4 levels in CRC metastases versus primary CRC or normal colon but these analyses are only of overall citrullination, without any analyses of ECM.

In Fig. 1 it was our intention to establish the nature of the proteomic platform that allowed us to evaluate the ECM of liver and liver metastases and uncover its modification by citrullination. The composition of the ECM in CRC is not the subject of the manuscript. Further analysis of this proteomic data revealed that many of the more abundant ECM proteins were more extensively citrullinated in the ECM of the metastases than in the ECM of the host tissue of the liver. As the reviewer points out, we then go on to

also evaluate the elements so identified, PAD4 and citrullination, in liver metastases, surrounding liver, primary colon carcinomas and their surrounding colon.

We fully agree that analysis of the ECM specifically was important and we now supply in Fig. 2G the measurement of citrullinated residues in all four tissues showing that the increased citrullination of ECM is significant and limited to the liver metastases.

The following sentence was added to the results: “The level of citrullinated peptides was also increased in ECM from CRC liver metastases compared to adjacent non-cancerous liver, primary CRCs and uninvolved colon (Fig. 2G).”

Furthermore, there are zero details about the sources of these colon and CRC samples.

The reviewer correctly points out that we omitted the source of some of the tissues. We have rectified this in the Materials and Methods section writing that “Human primary colorectal cancer and uninvolved colon from the same resection specimens and hepatic colorectal cancer metastasis tissue and surrounding unaffected liver tissue from resection specimens were obtained from patients from the Oxford Radcliffe Biobank following institutional review and granting of ethical approval (ethics number 09/H0606/5). Informed consent was obtained from all subjects involved.”

Unfortunately due to the nature of the consent under which the tissue bank operates, we are unable to obtain any additional clinical information other than the pathology of the individual cases.

Several CRC cell lines were also reported to show elevated levels of PAD4 and to secrete it into their culture medium (Figure 3). PAD4 is reported here to be associated with exosomes isolated from the culture medium (Fig. 3F-I), although there was no determination of the enrichment on isolation of the particles designated as exosomes and the identification of the particles as exosomes is limited.

In regard to the use of the word exosomes, we are aware that the definition of these particles is in flux. Hence we have changed the text to use extracellular vesicles (EVs) instead of exosomes to remove this issue from consideration. We provide further characterization of these EVs now showing that they express characteristic markers ALIX, TSG101, CD9, CD63, CD81 and lack expression of the cis-golgi marker GM130 and of the ER marker PDI. See Fig S8A. We also show enrichment of PAD4 in the EVs in Fig. S8B.

The manuscript text was altered in the following way: “In keeping with deposition of PAD4 through vesicle release, EVs isolated from cancer cell conditioned media (Fig. 3F) contained PAD4 as well as many characteristic EV markers including caveolin-1 and β -catenin and lacked markers of ER and of Golgi apparatus (Fig. 3G, Fig. S8A). PAD4 was enriched in EVs in the conditioned medium (Fig. S8B).”

PAD4 was also shown to be associated with ECM in the CRC liver mets (Fig 3J).

The effects of citrullination of collagen on cell behavior were then monitored; citrullination enhanced adhesion and reduced migration (Figure 4), although the effects on migration were marginal.

We now clarify in the text that “Cell motility, measured by median velocity and track length was also significantly decreased in colon cancer cells plated on citrullinated collagen type I compared to controls, albeit by only approximately 20-30%. “ to make the lesser extent of the mobility change clearer.

Some rather limited assays for integrin-mediated signal transduction events and for mRNA levels of markers for epithelial or mesenchymal state of the cells were then conducted (Fig. 4 E-G) and

interpreted as indicating that citrullination predisposed to an epithelial state. These analyses are insufficient to document the authors' conclusion about an MET.

We agree that this is insufficient. In addition to data presented in Fig. 4F and 4G, we have performed more PCR for selected epithelial and mesenchymal markers in LoVo cells plated onto citrullinated and intact collagen type I. See supplementary Fig. 10. The observed results were in line with previous findings showing a shift towards epithelial phenotype.

The manuscript text was altered in the following way: “Moreover, when plated on citrullinated collagen type I, the MC38 and LoVo colorectal cancer cells exhibited increased expression of epithelial markers (Fig. 4F, Fig. S10).”

Further, we generated CRC cells with overexpression of PAD4 to identify whether EMT markers are altered upon PAD4 overexpression. In tissue culture overexpression did not alter these markers (now added as Fig. S12D) consistent with the hypothesis that alteration in ECM citrullination is essential for this effect.

The following sentence was added to the Results section: “Of note, overexpression of PAD4 following transduction by an expression vector did not change the amounts of N- or E-cadherin or Slug (Fig. S12D).”

Finally, we have rewritten the whole manuscript text to state simply that the epithelial/mesenchymal markers have changed under the different conditions and only include MET in the discussion as a hypothesis. At the end of this response we provided several examples how the text has been changed as highlighted in **yellow**.

Mice injected with CRC cells were next treated or not with an inhibitor of PAD4 (BB-CI-amidine). The results were rather variable but in general showed reduced citrullination, although ECM proteins were not explicitly assayed.

Because of the small size of the treated metastases, we do not believe that it is technically possible for us to isolate the ECM from treated metastases or to obtain sufficient material to analyze it. This is because the methods we are using for ECM isolation and enrichment require a more substantial amount of tissue.

There was a reduction in metastatic burden on treatment with the PAD4 inhibitor and variable indications of reduced mesenchymal markers and increased epithelial markers but there is no evidence that this had anything to do with ECM proteins and shRNA knockdown of PAD4 did not affect EMT markers, although it did reduce metastases in two types of metastasis assays.

Our hypothesis is that citrullination of the ECM in liver metastases, not at subcutaneous or colonic sites is the crucial factor. We were able to show with the PAD inhibitor, BB-CI-amidine, that EMT markers changed in liver metastases. We could not ask the question after down-regulation in the intrasplenic injection model as liver metastases did not form at all (no material for analysis) (See Fig.5L, M, N) and in the spontaneous model the only liver metastases that formed had escaped from knockout of PAD4 (See Fig. 6G, H) also making this analysis not feasible.

However, the knockdown of PAD4 markedly reduced growth of subcutaneous primary tumors. This does not really fit well with the earlier claim that there is no change in citrullination in primary CRC.

We would point out here that whilst growth of 2 out of 3 cell lines was reduced, the growth of the third was not and that growth in the cecum was not reduced. We clearly indicated this in the manuscript.

The manuscript text was altered in the following way: “The cells with wild-type or PAD4 knockdown led to bowel tumors with similar weights in contrast to the decreased growth at the subcutaneous site (Fig. 6B).”

Again, these experiments need much more thorough analyses to be convincing – there are interesting hints of connections between citrullination, MET and metastasis but also inconsistencies that need to be followed up and explained.

Furthermore, there needs to be a better connection made between the multiple different strands of the paper; citrullination of ECM, exosomes, in vitro effects of citrullination of collagen, the MET hypothesis and the in vivo results – as discussed, the latter are not very reproducible or consistent. There are interesting preliminary results here but they need to be followed up in much more detail before publication could be recommended.

We thank the reviewer for their careful review of the manuscript. We believe that we have now performed all of the experiments suggested by this reviewer that were technically feasible and modified the text to ensure that hypothesis and experimental results are clearly distinguished. These experimental results together with those performed in response to reviewer 3 now make the connections apparent. That there is some variability in in vivo metastasis assays is not surprising and that some cell lines behave somewhat differently than others is also not unexpected. In each case, however, in several different models inhibition of PADs or knockdown of PAD4 led to dramatically limited liver metastasis.

Reviewer #2 (Remarks to the Author):

Overall, Yuzhalin et al show a novel role for ECM protein citrullination influences liver metastatic growth in colorectal cancer. This is a novel, and well controlled study. The authors should be congratulated on a comprehensive piece of work.

We thank the reviewer for appreciating our work.

Reviewer #3 (Remarks to the Author):

In this manuscript the Authors have analysed the extracellular matrix of the colon cancer, healthy colon, liver metastasis and healthy liver tissue by combining human and samples from mouse models and highly demanding technique of tissue decellularization. Their data indicate that extracellular matrix (ECM) of liver metastasis has increased levels of citrullinated proteins compared to all other tissues analysed. This posttranslational modification of proteins has been previously detected in autoimmune disease but the data in cancer is rather scarce thus underlying the novelty of this study. In vitro data suggest that the molecular mechanism that is controlled by this posttranslational modification of proteins may be the mesenchymal to epithelial transition (MET). In addition, in vivo data using PAD (peptidylarginine deiminase) inhibitor and three different colon cancer cell lines show similar results. However, there are some contradictory data when only PAD4 is inhibited that should be addressed before publishing as Authors state that PAD4 dependet citrullination of ECM drives liver metastasis formation. The manuscript is interesting, well written, with well performed experiments and provides novel data of potential interest.

Major points:

1. The use of PAD inhibitor in vivo produced the reduction of protein citrullination and reduction of metastasis formation using 3 different cell lines (LoVo, HCT116 and MC38). In addition, the Authors show the increase in expression of mesenchymal markers in metastatic lesions from mice treated with PAD inhibitor therefore implying that reduction of citrullination is producing the reversion of MET. However, when the expression of only PAD4 is downregulated using shRNAs, although there is a reduction in both the levels of extracellular and intracellular PAD4 as well as in levels of intracellular and extracellular citrullinated proteins no effect on the levels of expression of N-Cadherin and E-cadherin was observed. And still these cells were not able to form liver metastasis using two different cell lines. Therefore these results may imply that:

a) The effect seen on MET upon pharmacological inhibition of PAD is due to inhibition of other members of PAD family

b) PAD4 does not control liver metastasis formation through induction of MET

We believe that there is a third possibility that is consistent with these results. Our hypothesis is that citrullination of the ECM is the crucial factor, and this effect does not occur *in vitro* where ECM is sparse and collagen mainly absent (strictly speaking, there is no ECM *in vitro* apart from the one deposited by cells themselves). shRNA-transfected cells *in vitro* showed indeed no difference in N and E cadherins (Fig. 12E). However, when they were seeded onto citrullinated collagen type I (lanes 5 and 6 Fig.12E), we observed a change in N and E cadherins (suggesting that extracellular, not intracellular citrullination matters). Consistent with this hypothesis is the observation that plating CRC cells on collagen type I that has been citrullinated led to lesser EMT marker expression and altered attachment regardless of PAD4 downregulation (Figure 4B-G).

We have altered the text to make our hypothesis clearer (see below in yellow) and to more explicitly distinguish between the experimental results and the hypothesis. In general the text has been extensively rewritten throughout to make these points clear. The paragraphs indicated below are examples of our rewritten text to clearly state our hypothesis and to distinguish the hypothesis from the experimental results.

See below for examples of altered text.

For that reason Authors are suggested to either provide more evidence on PAD4 regulating MET or to separate citrullination effect on MET in liver metastasis formation from PAD4 mediated liver metastasis formation.

We have altered the manuscript throughout to distinguish between experimental results (alteration of EMT markers) and hypothesis (shift towards MET/EMT) clear. We agree with reviewer that we have not entirely proven this hypothesis, but only intend to propose it as a plausible explanation for the data.

Moreover, in addition to PAD4 overexpression analysis performed in response to reviewer 3 comment (Fig. S12D), we have performed more PCR for selected epithelial and mesenchymal markers in LoVo cells plated onto citrullinated and intact collagen type I. See supplementary Figure 10. The observed results were in line with previous findings showing a shift towards epithelial phenotype.

The manuscript text was altered in the following way: “Moreover, when plated on citrullinated collagen type I, the MC38 and LoVo colorectal cancer cells exhibited increased expression of epithelial markers (Fig. 4F, Fig. S10).”

2. Does overexpression of PAD4 affects the expression of epithelial and/or mesenchymal markers?

In response to this question, we generated CRC cells with overexpression of PAD4. In tissue culture overexpression did not alter these markers (now added as Fig. S12D) consistent with the hypothesis that alteration in ECM citrullination is essential for this effect.

The following sentence was added: “Of note, overexpression of PAD4 following transduction by an expression vector did not change the amounts of N- or E-cadherin or Slug (Fig. S12D).”

3. HCT116 cells when injected intraceum, beside primary tumor could develop both liver and lung metastasis. Is the formation of lung metastasis affected upon PAD4 downregulation in HCT116 cells?

We are aware of reports in the literature of intracecal tumors of HCT116 cells resulting in lung and other metastases as well as liver metastases. We examined all organs including the lung, but failed to find metastases in our experiments outside of the liver in any of the mice. This was clarified in the Figure 6 legends: “Other organs including lungs showed no evidence of metastasis”

Minor points:

1. In the figure 4A the presented data is according to figure legend from 2 independent experiments. However, error bars were not provided. Please provide this data. In the same figure the effect of PAD inhibitor BB-CI-amidine on the amount of citrullinated peptides is bigger when 20 times lower dose of inhibitor is used (see data on RhColI +PAD4 +1mM BB vs RhColI +PAD4 +0.05mM BB). How the Authors explain this?

Error bars are now provided. There is a small difference in the effect of the doses of the BB-CI-amidine which we attribute to experimental variation.

2. Please provide better resolution images for the Figures 5C and 6D

Higher magnification images have been substituted in these figures.

3. The Authors claim that GW4869 inhibitor was not tested on HCT116 and LoVo cells due to toxicity. However, the data presented in the figure S6E do not show the effects of toxicity rather the reduction of proliferation. Please comment on this data.

We believe the issue here is the word toxicity which we took to mean any deleterious effect including reduction of proliferation. Further the WST-1 assay does not distinguish between viability and proliferation so that we have changed that text to read the following: “HT29 but not HCT116 and LoVo, was unaffected by GW4869 in a WST-1 assay of cell proliferation and viability (Fig. S8C). Hence EVs were only analyzed from HT29.”

4. Please provide the details in the Figure legend for all the blots presented in the figure S7B.

We have added arrows in the figure that link the identification of the cell lines between Fig. S7A and B. The legend now notes “Large arrows show corresponding data for each cell line in A and B.”

5. Please provide the explanation why in some blots of citrullinated proteins almost all molecular weights (e.g. S10B) of proteins are presented whereas in some blots only low molecular weight

proteins are shown (e.g. Figures 5A and B)? Is BB inhibitor affecting citrullination of only low molecular weight proteins?

In our experience, the blots of proteins separated by gel electrophoresis and then treated using reagents that react with citrulline residues followed by immunoblotting do not effectively distinguish individual bands except at lower molecular weights. At the higher molecular weights this results in smears of different intensity. Below, we show the originals from Fig. 5A and B. We cropped them in the figure as an esthetic choice. As you can see inhibition of citrullination by BB-CI-amidine affects all sized proteins. Fig S10B has less smear because it is from in vitro material rather than from actual tissues. We thank the reviewer for appreciating our work.

The text has been changed throughout in avoiding the use of MET term, only suggesting it as a hypothesis. Specific examples of the text stating the hypothesis are as follows:

Page 5, paragraph 3

In particular, RNA expression of mesenchymal markers N-cadherin (*CDH2*) and Snail1 and 3 (*SNAI1* and 3) were reduced in MC38 cells plated onto citrullinated collagen, whilst expression levels of E-cadherin (*CDH1*) and other epithelial markers were increased. Reciprocal changes in N- and E-cadherin were confirmed at the protein level by immunoblotting (Fig. 4G). **Thus, adhesion to citrullinated collagen type I promoted alterations in EMT markers in CRC cell lines.**

Page 6, paragraph 3

We then asked whether downregulation of PAD4 in CRC cells had analogous effects to pharmacological inhibition. PAD4 levels were reduced by introduction of the shRNA in HT29 and HCT116 cells (Fig. S12A) and resulted in decreased intracellular and secreted protein citrullination (Fig. S12B). In tissue

culture expression of the EMT markers N- and E-cadherin was unchanged suggesting that intracellular PAD4 and intracellular citrullination have minimal effects on epithelial-mesenchymal plasticity in these cells (Fig. S12C). This is in striking contrast to the shift in EMT markers observed after plating the cells with downregulated PAD4 on citrullinated collagen type I (Fig. S12C) suggesting that engagement with citrullinated ECM rather than intracellular PAD4 drove the changes in EMT characteristics. Of note, overexpression of PAD4 following transduction by an expression vector did not change the amounts of N- or E-cadherin or Slug (Fig. S12D).

Page 7, paragraph 3

We found PAD4 at higher levels in human CRC liver metastases than in primary CRC, or adjacent colon or liver. Consistent with the enzymatic activity of PAD4, citrullinated residues were present at higher levels in both total lysates and in the ECM of liver metastatic tissues than in primary CRC, liver or colon. Our evidence suggests that extracellular PAD4 possibly through citrullination of collagen type I and other ECM proteins alters the characteristics of CRC cells. Downregulation of PAD4 in tissue culture, a situation in which ECM is sparse did not alter EMT characteristics. However plating cells on citrullinated collagen altered their EMT features regardless of intracellular PAD4. In experimental liver metastasis models, downregulation or pharmacological inhibition of PAD4 reduced citrullination and liver metastatic growth. Taken together, these experiments suggest that the citrullination of ECM molecules requiring extracellular PAD4 could be an important event in altering CRC cell signaling during metastatic growth. Unlike the CRC cells used here, downregulation of PAD4 in breast²⁹ and lung³⁰ cancer cells in tissue culture resulted in decreased EMT markers. Thus, signaling to reduce EMT marker expression may be mediated both intra- and extracellularly, reinforcing the possibility that PAD4 and citrullination might be a therapeutic target in liver metastases.

Page 8, paragraph 2

Other molecules we identified in the ECM of metastatic liver also have the capacity to affect EMT. Versican, which was considerably overexpressed in liver metastasis ECM, has been reported to induce MET³⁶. TGF- β signaling has been recognized as critical to CRC liver metastasis³⁷ and we found upregulation of the TGF- β binding proteins LTBP1-3, TSP1 and TSP2, which contribute to TGF- β activation^{38,39}. TGF- β has been linked to tumor suppression through EMT, and its citrullination altered its activity^{39,40}. The complex balance between these signals and the function of epithelial-mesenchymal plasticity in cancer is still not completely understood. Genetic manipulation of EMT in murine cancer models leads to increased invasion and migration by metastatic cells⁴¹ whilst more recent work implicates EMT in promotion of tumor-initiating cells and resistance to chemotherapy^{42,43}. After dissemination, MET has been suggested to be necessary to initiate metastatic colony formation⁴⁴⁻⁴⁶. Further, MET has been documented in human CRC liver metastases as demonstrated by Hur et al.⁴⁷, showing markers for MET in liver metastases compared to their matched primary lesions. This is consistent with our data and the hypothesis that cancer cell secretion of PAD4 leads to MET due to modification of the metastatic liver ECM by citrullination with the end result of facilitating metastatic growth. This hypothesis points to cancer cell plasticity being enforced by an extracellular signal, that of citrullination of ECM components. This novel mechanism could be exploited by cancer cells to secure their growth at distant sites, and interruption of this signaling could generate an unfavorable microenvironment for cancer cell growth. Therefore, PAD4 and citrullination might be a promising target in therapy of liver metastases.

Reviewers' comments:

Reviewer #2 (Remarks to the Author):

The additional details provided by the reviewers in this revised manuscript further support their original conclusions, and I have not other requests.

Reviewer #3 (Remarks to the Author):

The Authors have addressed or provided reasonable explanations to all the issues raised by this reviewer. The new version of manuscript that incorporates new data represents a significant improvement. In addition, text re-writing represents a notable effort to clarify the raised. To this reviewer, the new data provides sufficient evidence to sustain that PAD4 is implicated in colon cancer liver metastasis formation. In addition, other experiments provided also clearly indicates that citrullination controls liver metastasis formation through the control of expression of epithelial and mesenchymal markers. However, whether ECM citrullination is totally driven by PAD4 and whether this is the only role that PAD4 has in liver metastasis could still be a cause of debate.

In summary, taking into account all presented data and the eagerness of the field to understand the basis of metastasis, the current manuscripts represents a significant effort in trying to decipher process of liver colonization in colon cancer and thus I would recommend the manuscript for publication.

Reviewer #4 (Remarks to the Author):

This study represents a large amount of work with multiple human CRC cell lines, and a murine CRC line, in proteomics, in vitro assays and xenograft studies of sub-cutaneous tumor growth and a liver metastasis (spleen injection) model in mice.

The authors study the role of PAD4 in driving citrullination of proteins and how PAD4 is necessary (via an inhibitor or bsirna-silencing of PAD4) for liver metastasis of xenografted CRC cells in mice. It is shown that PAD4 makes a major contribution to citrullination of ECM proteins and that collagen I citrullination can affect the status of epithelial versus mesenchymal markers in the CRC lines. This is linked to the in vivo context in PAD4 inhibition experiments (but not in the PAD4 siRNA design). The data show that PAD4 is needed for liver metastasis in these models, which is interesting, but the mechanistic aspects have not been brought together as yet.

A major reservation for this reviewer is the inappropriate use of t-tests throughout. Some claims may not be justified.

There is inconsistency of PAD4 bands (molecular weights and the total number of bands) in many of the blots, leaving some experiments unconvincing.

There are 12 complex supplementary figures and some of these are used to present results which should really be part of the main figures. The referencing to figures moves between main and supplementary figures, making the logic of the results difficult to follow. Particularly S11 and S12 present data that are central to the advances made by the study that should be incorporated in the main figures.

Specific comments.

1. A major issue that applies throughout the manuscript is that, with the exception of time-lapse dataset, the t-test has been used to test for statistical significance. This does not appear the appropriate test. In experiments where 2 conditions were compared, the scatter points show that

variability was non-uniform between the two conditions. Therefore the non-parametric Mann-Whitney U test would be the appropriate test. In relation to this, it is doubted that Fig. 6F reaches statistical significance by Mann-Whitney test.

Where more than two conditions are compared, eg Fig. 2E and many others which show unequal variance between the expt. conditions, a non-parametric test such as the Kruskal-Wallis would be appropriate. As currently presented, it is likely that statistical significance is over-estimated (for example 2E, 2F, where the range of variation in the liver mets. condition overlaps with the datapoints other conditions). So it is not clear what results can be deemed to be of statistical significance. I have used Fig 2 for examples, but this issue applies to almost all figures.

2.P3, paragraph 2: please also state what % of the identified proteins had signal peptides, and what % also lacked TM domains and so could be secreted.

3.PAD4 blots. The various panels show variability beyond the 74kDa and dimer bands mentioned in the text. For example, Fig S6C has many bands: antibody specificity and reliability is not convincing. Fig S6D shows another band pattern in some cells. In panels of Fig 3 related to conditioned media, it is inconsistent whether the band presents at 74kda or the dimer – why? Fig S12D also shows multiple PAD4 bands.

4.In Fig. 3E, how were bands visible by Coomassie stain obtained from conditioned media? This appears an unusual result.

5.In Fig 3J and S8D the distinction between intracellular and extracellular staining is not convincing.

6.Fig 4B: the cell attachment assays are carried out with single conditions of the standard collagen I or citrullinated preparations. To demonstrate differential cell attachment effectively, a dose-response experiment would be needed. Also in relation to the design of this experiment: was any citrullination detected in the standard collagen prep ? (by blot). What was the level of increase achieved by the in vitro PAD4 treatment? And is the increase within the physiological range? Since cells exposed to collagen + PAD4 attached and moved less and have less active signalling, the viability and /or apoptosis of these cells also need to be tested in comparison to the control conditions to rule out that the modified collagen simply affects viability.

7.The effect of the integrin blocking antibodies under control conditions appears modest and is not in line with published results on roles of alpha 1 and alpha2 integrins in collagen-binding.

8.P6 in general: Many results are presented in Figs S11 and S12 that are essential to understand the advances made and the ensuing model. It would be preferable to put these in the main figures. In general, the flow from main figures to supplementary figures and vice versa is difficult to follow.

Minor.

Some points of wording lead to over-statements.

1.Line 82/83, Fig. S1B: the decellularized liver scaffolds are stated to “consist predominantly” of ECM proteins, but no non-ECM proteins were tested for. Suggest revise wording.

Line 84, re S1C: since the above point is not correct, the “increased proportion of high mol. Wt. proteins” does not “confirm”. “Support” would be more appropriate.

Line 263, Fig 5K, the decreased growth of HCT116 tumours is apparent only at one timepoint, please make a more cautious statement.

2.Graph presentation : for 5-6 datapoints, a box and whisker plot is not really needed.

3.Line 159: transcripts are measured in Fig. 3C, so “expressed” would be more suitable than “produced”

4.Lines 281/282, this sentence states “.. concurrently altering EMT markers”, however EMT

markers have not been examined in vivo in the experiment with siRNA knockdown of PAD4.

Reviewers' comments:

Reviewer #2 (Remarks to the Author):

The additional details provided by the reviewers in this revised manuscript further support their original conclusions, and I have not other requests.

We thank the reviewer for their careful assessment of the manuscript

Reviewer #3 (Remarks to the Author):

The Authors have addressed or provided reasonable explanations to all the issues raised by this reviewer. The new version of manuscript that incorporates new data represents a significant improvement. In addition, text re-writing represents a notable effort to clarify the raised. To this reviewer, the new data provides sufficient evidence to sustain that PAD4 is implicated in colon cancer liver metastasis formation. In addition, other experiments provided also clearly indicates that citrullination controls liver metastasis formation through the control of expression of epithelial and mesenchymal markers. However, whether ECM citrullination is totally driven by PAD4 and whether this is the only role that PAD4 has in liver metastasis could still be a cause of debate.

In summary, taking into account all presented data and the eagerness of the field to understand the basis of metastasis, the current manuscripts represents a significant effort in trying to decipher process of liver colonization in colon cancer and thus I would recommend the manuscript for publication.

We thank the reviewer for their careful assessment of the manuscript

Reviewer #4 (Remarks to the Author):

This study represents a large amount of work with multiple human CRC cell lines, and a murine CRC line, in proteomics, in vitro assays and xenograft studies of sub-cutaneous tumor growth and a liver metastasis (spleen injection) model in mice.

The authors study the role of PAD4 in driving citrullination of proteins and how PAD4 is necessary (via an inhibitor or bsirna-silencing of PAD4) for liver metastasis of xenografted CRC cells in mice. It is shown that PAD4 makes a major contribution to citrullination of ECM proteins and that collagen I citrullination can affect the status of epithelial versus mesenchymal markers in the CRC lines. This is linked to the in vivo context in PAD4 inhibition experiments (but not in the PAD4 siRNA design). The data show that PAD4 is needed for liver metastasis in these models, which is interesting, but the mechanistic aspects have not been brought together as yet.

A major reservation for this reviewer is the inappropriate use of t-tests throughout. Some claims may not be justified.

There is inconsistency of PAD4 bands (molecular weights and the total number of bands) in many of the blots, leaving some experiments unconvincing.

There are 12 complex supplementary figures and some of these are used to present results which should really be part of the main figures. The referencing to figures moves between

main and supplementary figures, making the logic of the results difficult to follow. Particularly S11 and S12 present data that are central to the advances made by the study that should be incorporated in the main figures.

Specific comments.

1.A major issue that applies throughout the manuscript is that, with the exception of time-lapse dataset, the t-test has been used to test for statistical significance. This does not appear the appropriate test. In experiments where 2 conditions were compared, the scatter points show that variability was non-uniform between the two conditions. Therefore the non-parametric Mann-Whitney U test would be the appropriate test. In relation to this, it is doubted that Fig. 6F reaches statistical significance by Mann-Whitney test.

Where more than two conditions are compared, eg Fig. 2E and many others which show unequal variance between the expt. conditions, a non-parametric test such as the Kruskal-Wallis would be appropriate. As currently presented, it is likely that statistical significance is over-estimated (for example 2E, 2F, where the range of variation in the liver mets. condition overlaps with the datapoints other conditions). So it is not clear what results can be deemed to be of statistical significance. I have used Fig 2 for examples, but this issue applies to almost all figures.

We have no disagreements with the above comments regarding statistical assessments. With N numbers less than 15-20 it is well recognized to be difficult to adequately evaluate the normality of distribution, therefore both parametric and nonparametric tests are often considered acceptable. Taking into account the reservations raised by the reviewer, we have revised the manuscript using mainly non-parametric statistical tests. In Summary:

- For analysis of 2 groups with **unpaired** samples, we now use the Mann-Whitney U test;
- For analysis of 2 groups with **paired** samples, we now use the Wilcoxon signed rank test;
- For analysis of 3 or more groups, we now use the Kruskal-Wallis test with Dunn's multiple comparison post-test;
- In Fig. 2B, values in both groups were normally distributed (based on the Kolmogorov-Smirnov test: $P > 0.1$). Therefore we used the paired Student's T test.

After implementation of these changes using the usual guideline of $p \leq 0.05$, the experiments retain statistical significance with one exception. That is of the Elisa of PAD4 in human tissues in Fig 3A. The Western blot analysis of those tissues and difference achieved by LC-MS/MS analysis however retains significance. Overall this does not alter the results or conclusions of the article.

2.P3, paragraph 2: please also state what % of the identified proteins had signal peptides, and what % also lacked TM domains and so could be secreted.

Estimation of the total number of secreted proteins in a LC-MS/MS dataset is difficult, and can be done in a variety of ways. Gene ontology analysis of our dataset (not shown) identifies 12.5% of hits as extracellular matrix proteins, which clearly can be classified as secreted. This is a minimum estimation as it excludes other proteins that could be secreted as well. Iterative screening for signal peptides and TM domains requires the use of some quite

sophisticated bioinformatics pipelines. Consideration of the means of access to the ECM of all proteins in that compartment seems beyond the scope of this study.

3. PAD4 blots. The various panels show variability beyond the 74kDa and dimer bands mentioned in the text. For example, Fig S6C has many bands: antibody specificity and reliability is not convincing. Fig S6D shows another band pattern in some cells. In panels of Fig 3 related to conditioned media, it is inconsistent whether the band presents at 74kDa or the dimer – why? Fig S12D also shows multiple PAD4 bands.

As the reviewer notes in addition to the more prominent bands of the 74 kDa monomer and the 148 kDa dimer some intermediate bands are also noted. As we pointed out in the text PAD4 can be observed as a monomer or a dimer which varies with Ca^{++} concentration. The evidence that these bands all indicate PAD4 is that downregulation of PAD4 genetically by shRNA greatly reduces these bands (Fig 7) and overexpression of PAD4 enhances their intensity (Fig. S12). Further evidence that the antibody recognizes PAD4 is that it reacts with recombinant PAD4 in immunoblotting (Fig S6). We have added a sentence to the text to add these points.

“The dimeric form was generally more abundant in our samples (Fig. S6A, B). Evidence that the major bands represent PAD4 is that genetic downregulation greatly reduced them and overexpression enhanced them (see Fig 7A and Fig. S12B). In addition more minor intermediate bands, which could include degradation products, other modifications or degradation products were seen. These bands were enhanced after overexpression (Fig S12B). Further the antibody shows specificity in that it reacts with recombinant PAD4 (see Fig. S6).”

Overall, we recognise that it is very difficult to blot for PAD4. In our lab we tried 4 different commercial antibodies for PAD4 and the antibody we used in these studies produced the most reliable results. Therefore, we do not believe there are means of improving western blotting quality at this point.

4. In Fig. 3E, how were bands visible by Coomassie stain obtained from conditioned media? This appears an unusual result.

We apologize for this confusion. In the experiment, original media samples were first concentrated using special columns as described in the Methods section, then split in two parts and loaded into two parallel gels. One gel was blotted for PAD4, and another one was Coomassie stained (see below). We have slightly altered the Figure to clearly indicate that one panel is the western blot and another one is a Coomassie stained gel.

We performed this experiment this way, because there is currently no well recognised and ubiquitously expressed loading controls for conditioned media. Coomassie staining is frequently used as a loading control.

PAD4

Coomassie

5. In Fig 3J and S8D the distinction between intracellular and extracellular staining is not convincing.

We co-stained tissues for PAD4 and for the extracellularly deposited proteins Collagen I and IV. Co-localisation with the collagens therefore implies the extracellular location of PAD4, whereas perinuclear PAD4 staining suggested intracellular localisation (see below). We have slightly enlarged the image in the manuscript to make it clearer.

PAD4 extracellular expression

PAD4 intracellular expression

6.Fig 4B: the cell attachment assays are carried out with single conditions of the standard collagen I or citrullinated preparations. To demonstrate differential cell attachment effectively, a dose-response experiment would be needed. Also in relation to the design of this experiment: was any citrullination detected in the standard collagen prep ? (by blot). What was the level of increase achieved by the in vitro PAD4 treatment? And is the increase within the physiological range? Since cells exposed to collagen + PAD4 attached and moved less and have less active signalling, the viability and /or apoptosis of these cells also need to be tested in comparison to the control conditions to rule out that the modified collagen simply affects viability.

In the original manuscript and in this revision, we provided a mass spectrometry analysis of in vitro citrullinated collagen. The level of increase achieved by the in vitro PAD4 treatment is approximately 10-fold.

To address the reviewer`s concerns, we have performed an additional western blot analysis. Collagen type I pre-treated with recombinant PAD4 (ascending concentration) or collagen I pre-treated with recombinant PAD4 and BB-CI-amidine (ascending concentration) were subjected to immunoblotting for citrullinated proteins. Negative controls were recombinant collagen type I alone or depletion of Ca²⁺ in the experimental group. As seen, ascending

concentrations of PAD4 promote citrullination of collagen I in vitro using this assay. This data is now shown in Supplementary Fig 9A.

Both of these assays would suggest a low baseline level of citrullination. It is hard to speculate on whether or not this increase in citrullination is within the physiological range as there is essentially no literature on this point. A more detailed analysis of how much citrullination occurs in physiological or pathological specimens and how that might affect attachment or other features is beyond the scope of this manuscript which as noted by the reviewer already contains a substantial amount of experimentation. Mapping the many sites that are citrullinated even within collagen type I alone and determining the amount of citrullination at each and the effects of citrullination at each would be a separate study in itself.

We agree with the reviewer that viability of cells could be affected by seeding them on citrullinated collagen. We would point out that while motility decreased on the citrullinated collagen, attachment increased so that loss of viability might not be a simple explanation. To address this issue, we have performed an additional experiment of culturing 4 CRC cell lines on unmodified and citrullinated collagen (with ascending concentration of PAD4). Proliferation and viability were examined at time points 24, 48, 72, and 96 h. As seen from Fig S9B,C and below, there was no significant alteration of proliferation or viability of cells seeded on citrullinated collagen. Therefore, we do not believe citrullination reduced viability to a detectable degree in this setting.

7. The effect of the integrin blocking antibodies under control conditions appears modest and is not in line with published results on roles of alpha 1 and alpha 2 integrins in collagen-binding.

We agree with the reviewer that the observed results are possibly not in line with the published data. We have removed the experimental group with integrin-blocking antibodies from the panel. We will address this issue specifically in future studies.

8.P6 in general: Many results are presented in Figs S11 and S12 that are essential to understand the advances made and the ensuing model. It would be preferable to put these in the main figures. In general, the flow from main figures to supplementary figures and vice versa is difficult to follow.

We agree with the reviewer. We have rearranged the Figures, so currently there are 8 main Figures and 12 supplementary Figures.

Minor.

Some points of wording lead to over-statements.

1. Line 82/83, Fig. S1B: the decellularized liver scaffolds are stated to “consist predominantly” of ECM proteins, but no non-ECM proteins were tested for. Suggest revise wording.

We have reworded that. We changed “consist predominantly” to “contained”.

Line 84, re S1C: since the above point is not correct, the “increased proportion of high mol. Wt. proteins” does not “confirm”. “Support” would be more appropriate.

We have changed “confirm” to “consistent with”.

Line 263, Fig 5K, the decreased growth of HCT116 tumours is apparent only at one timepoint, please make a more cautious statement.

We have change the appropriate sentence to indicate that.

2. Graph presentation : for 5-6 datapoints, a box and whisker plot is not really needed.

Editorial Policy checklist for Nature Communications requires “to “present data in a format that shows data distribution”. We believe dot plots combined with box and whiskers are very informative as compared to bars. We only used box and whiskers in analyses where normality of distribution cannot be adequately assessed due to low sample size in order to give a better impression of difference change.

3. Line 159: transcripts are measured in Fig. 3C, so “expressed” would be more suitable than “produced”

We have changed “produced” to “expressed”

4. Lines 281/282, this sentence states “.. concurrently altering EMT markers”, however EMT markers have not been examined in vivo in the experiment with siRNA knockdown of PAD4.

We have deleted this statement.

We thank the reviewer for the careful and helpful assessment of the manuscript.

REVIEWERS' COMMENTS:

Reviewer #4 (Remarks to the Author):

The authors are thanked for addressing the majority of the points raised by this reviewer and for improving the balance between main figures and supplementary figures. A couple of items remain.

1. With regard to the immunofluorescence of intracellular/extracellular PAD4, this reviewer remains unconvinced by the presented XY images. XZ sections, 3D projection in conjunction with staining with a plasma membrane marker would be needed. Without this clarity, please qualify the wording.
2. The meaning of my comment on the box and whisker plots is that, for the number of datapoints presented, the box and whiskers are not needed. The scatter data points can be presented clearly with mean bar \pm SD / or median bar \pm 95% confidence interval (e.g., as in Fig. 4B).

REVIEWERS' COMMENTS:

Reviewer #4 (Remarks to the Author):

The authors are thanked for addressing the majority of the points raised by this reviewer and for improving the balance between main figures and supplementary figures. A couple of items remain.

1. With regard to the immunofluorescence of intracellular/extracellular PAD4, this reviewer remains unconvinced by the presented XY images. XZ sections, 3D projection in conjunction with staining with a plasma membrane marker would be needed. Without this clarity, please qualify the wording.

In this experiment we used epifluorescent, not confocal, microscopy, and therefore Z-stacks were unavailable. As we showed co-localization of collagen staining with PAD4 staining, the extracellular location of PAD4 does not seem to be in contention. The issue here would seem to be whether PAD4 could be identified as cellular as well, which is not a key point. While it is likely that the PAD4 staining in areas identified as cellular do indeed represent cellular content, we agree that the evidence is less than direct. We have therefore reworded the sentence in the following way to match the text to the images:

“Finally, immunohistochemistry of human CRC liver metastases revealed extracellular PAD4 localization. The extracellular PAD4 colocalized to collagens type I and IV providing additional evidence for the presence of PAD4 in the tumor ECM (Fig. 3J and Supplementary

Fig.8D). Additional PAD4 was associated with cellular areas.”

2. The meaning of my comment on the box and whisker plots is that, for the number of datapoints presented, the box and whiskers are not needed. The scatter data points can be presented clearly with mean bar \pm SD / or median bar \pm 95% confidence interval (e.g., as in Fig. 4B).

As it was mentioned by the reviewer, for 5-6 datapoints the box and whiskers plot is not needed. We therefore changed the box and whiskers plots in the entire Figure 8 for scatter dot plots.